# Control of plant cell fate transitions by transcriptional and hormonal signals

Christophe Gaillochet[1], Thomas Stiehl[2,3†], Christian Wenzl[1†], Juan-José Ripoll[4], Lindsay J Bailey-Steinitz[4], Lanxin Li[1], Anne Pfeiffer[1], Andrej Miotk[1], Jana P Hakenjos[1], Joachim Forner[1‡], Martin F Yanofsky[4], Anna Marciniak-Czochra[2,3,5], Jan U Lohmann[1*]

[1]Department of Stem Cell Biology, Centre for Organismal Studies, University of Heidelberg, Heidelberg, Germany; [2]Institute of Applied Mathematics, Heidelberg University, Heidelberg, Germany; [3]Interdisciplinary Center for Scientific Computing, Heidelberg University, Heidelberg, Germany; [4]Division of Biological Sciences, Section of Cell and Developmental Biology, University of California, San Diego, San Diego, United States; [5]Bioquant Center, Heidelberg University, Heidelberg, Germany

**\*For correspondence:**
jan.lohmann@cos.uni-heidelberg.de

[†]These authors contributed equally to this work

**Present address:** [‡]Max-Planck-Institut für Molekulare Pflanzenphysiologie, Potsdam, Germany

**Competing interests:** The authors declare that no competing interests exist.

**Abstract** Plant meristems carry pools of continuously active stem cells, whose activity is controlled by developmental and environmental signals. After stem cell division, daughter cells that exit the stem cell domain acquire transit amplifying cell identity before they are incorporated into organs and differentiate. In this study, we used an integrated approach to elucidate the role of *HECATE* (*HEC*) genes in regulating developmental trajectories of shoot stem cells in *Arabidopsis thaliana*. Our work reveals that *HEC* function stabilizes cell fate in distinct zones of the shoot meristem thereby controlling the spatio-temporal dynamics of stem cell differentiation. Importantly, this activity is concomitant with the local modulation of cellular responses to cytokinin and auxin, two key phytohormones regulating cell behaviour. Mechanistically, we show that HEC factors transcriptionally control and physically interact with MONOPTEROS (MP), a key regulator of auxin signalling, and modulate the autocatalytic stabilization of auxin signalling output.
DOI: https://doi.org/10.7554/eLife.30135.001

## Introduction

The evolutionary success of multicellular organisms is based on the diversification of cellular identities and the division of labour among cell types. To orchestrate this diversity, complex signalling systems have evolved to guide stem cell differentiation based on hard-wired developmental programs and environmental signals (reviewed in *[Pfeiffer et al., 2017]*). Plants represent particularly attractive models to study the molecular mechanisms underlying the transition from stem cell to differentiated cell fate: Firstly, plants employ a postembryonic mode of development, which is based on the continuous activity of pluripotent stem cells embedded in specialized tissues, called meristems. Secondly, plant development is modular and thus the same set of organs is initiated repeatedly from a stem cell system, greatly facilitating in vivo analysis of cell-decision-making. Thirdly, due to the encasement by a cell wall, plant cells are immobile and thus their identity is determined by position, rather than lineage and can change multiple times during their development until terminal differentiation.

In the shoot apical meristem (SAM), the stem cell system responsible for the generation of all above ground structures, two major fate transitions can be identified: From stem cells in the central zone (CZ) to transit amplifying cells in the peripheral zone (PZ) and further on into organ primordia, which will give rise to fully differentiated lateral structures, such as leaves or flowers (reviewed in

**eLife digest** Unlike animals, plants continuously generate new organs that make up their body. At the core of this amazing capacity lie tissues called meristems, which are found at the growing tips of all plants. Meristems contain dividing stem cells. The daughters of these stem cells pass through nearby regions called transition domains. Over time, they change – or differentiate – to go on to become part of tissues like leaves, roots, stems, shoots, flowers or fruits.

Stem cell differentiation has a direct impact on a plant's architecture and eventually its reproductive success. For crops, these factors determine yield. This means that understanding this aspect of plant development is central to basic and applied plant biology. Many factors required for shoot meristem activity have been identified, with a focus so far on the processes that control the identity of the cells produced.

Now, Gaillochet et al. have asked which genes are responsible for controlling when stem cells in meristems differentiate. The analysis focused on the meristem that makes all the above ground parts of model plant *Arabidopsis thaliana* – the shoot apical meristem. Gaillochet et al. found that *HECATE* genes (or *HEC* for short) control the timing of stem cell differentiation by regulating the balance between the activities of two plant hormones: cytokinin and auxin. These genes promote cytokinin signals at the centre of the meristem, and dampen auxin response at the edges. This acts to slow down cell differentiation in two key transition domains of the shoot meristem.

These new findings provide a molecular framework that now can be further investigated in crop plants to try to improve their yield. The findings also lay the foundation for studies of animals that may define common principles shared among stem cell systems in organisms that diverged over a billion years ago.

DOI: https://doi.org/10.7554/eLife.30135.002

[*Gaillochet et al., 2015*]). At the molecular level, cell fate trajectories are instructed by an intertwined communication system between local transcriptional networks and non-cell autonomous phytohormone signals (*Brand et al., 2000*; *Gordon et al., 2009*; *Jasinski et al., 2005*; *Leibfried et al., 2005*; *Schoof et al., 2000*). Stem cell fate in the SAM is dependent on the homeodomain transcription factor WUSCHEL (WUS), whose RNA is expressed in the organising centre (OC), located below the stem cells. WUS protein moves apically through plasmodesmata into the overlying cells, where it is required to maintain stem cell identity (*Daum et al., 2014*; *Yadav et al., 2011*). Stem cells in turn express CLAVATA3 (CLV3), a short, secreted peptide that acts to limit *WUS* expression via the CLV1, CLV2, CORYNE (CRN), BARELY ANY MERISTEM (BAM) receptors system (*Bleckmann et al., 2010*; *Clark et al., 1997*; *Fletcher et al., 1999*; *Nimchuk et al., 2015*; *Ohyama et al., 2009*). The resulting negative feedback loop represents the core module of SAM regulation and couples the size of the OC with that of the CZ (*Brand et al., 2000*; *Schoof et al., 2000*). In parallel, the KNOTTED-like homeobox transcription factor SHOOT MERISTEMLESS (STM) is required throughout the SAM to inhibit differentiation by stimulation of cytokinin production and repression of gibberellic acid (GA) biosynthesis (*Jasinski et al., 2005*; *Long et al., 1996*). Transcriptional and hormonal regulation interact to control SAM activity in spatially distinct sub-domains: In the centre of the SAM, WUS represses A-Type *ARABIDOPSIS RESPONSE REGULATOR* (*ARR*) genes, which encode for negative feedback factors in cytokinin signalling. In essence, WUS acts to sensitize the cellular environment to cytokinin, which in turn promotes *WUS* expression thus creating a positive feedback loop that helps to establish and maintain the OC (*Gordon et al., 2009*; *Leibfried et al., 2005*). Cytokinin also plays a key role in controlling cell proliferation in the SAM by stimulating *CYCLIN D3* expression (*Riou-Khamlichi et al., 1999*).

In addition to its role in integrating hormonal signals, WUS directly represses the expression of *HECATE1* (*HEC1*), which encodes a bHLH transcription factor that redundantly functions with its closest paralogs *HEC2* and *HEC3* in various developmental contexts (*Gremski et al., 2007*; *Schuster et al., 2015*; *Schuster et al., 2014*; *Zhu et al., 2016*). In line with this regulation, *HEC1* mRNA is expressed throughout the SAM, but excluded from the OC (*Figure 2—figure supplement 1*; *Schuster et al., 2014*). This pattern is faithfully translated into protein accumulation, since HEC1 protein displays limited intercellular movement (*Daum et al., 2014*; *Schuster et al., 2014*).

Importantly, the precise spatial control of *HEC1* activity by WUS is essential for SAM function, since uncoupling *HEC1* expression from WUS in the OC leads to SAM termination, whereas enhancing *HEC1* activity in stem cells leads to massive over-accumulation of these cells followed by a progressive repression of the core WUS/CLV3 feedback system (*Schuster et al., 2014*). Mechanistically, this function is mediated by the formation of a protein complex between HEC and the bHLH transcription factor SPATULA (SPT) (*Schuster et al., 2014*).

In contrast to the long term maintenance of stem cells in the centre of the SAM, lateral organ primordia are continuously initiated at the periphery and cells are guided towards differentiation in restricted domains defined by the accumulation of auxin (reviewed in [*Weijers and Wagner, 2016*]). Local auxin signalling maxima are dynamically formed through a combination of different mechanisms that include auxin biosynthesis and the controlled intracellular polarization of the auxin transporter PIN-FORMED1 (PIN1) by the activity of the protein kinase PINOID (PID) (*Benjamins et al., 2001*; *Pinon et al., 2013*; *Reinhardt et al., 2003*). Dynamic auxin maxima are translated into robust auxin signalling output by AUXIN RESPONSE FACTOR transcription factors, such as ARF5/MONOPTEROS (ARF5/MP) (*Bhatia et al., 2016*; *Hardtke and Berleth, 1998*). MP associates with the chromatin remodelling factors SPLAYED (SYD) and BRAHMA (BRM), forming a regulatory protein complex sufficient to promote floral fate over undifferentiated SAM fate (*Wu et al., 2015*). In addition to primordia initiation, a boundary zone (BZ), which surrounds the entire SAM like a ring is required for proper spatial separation of organs from the SAM (reviewed in [*Žádníková and Simon, 2014*]). The BZ is dependent on the activity of the *CUP SHAPED COTYLEDONS* (*CUC*) and *LATERAL ORGANFUSION* (*LOF*) genes and loss of their function results in extensive fusions of lateral organs with the active SAM (*Aida et al., 1999*; *Gaillochet et al., 2015*).

Importantly, the activity of the central domain of the SAM is tightly coupled with the specification of lateral organs at the periphery by inter-domain communication systems. First, MP relays auxin signals to the core *WUS/CLV3* feedback loop by negatively regulating *ARR7* (*Zhao et al., 2010*). Second, organ primordia produce the short peptide CLE27, which signals in the centre of the SAM through AtFEA3 and represses *WUS* expression (*Je et al., 2016*). As a consequence of these multiple regulatory loops, the rate of lateral organ initiation correlates with the size of the SAM and larger SAMs tend to produce more organs per unit of time (*Landrein et al., 2015*).

Given the intertwined nature of the regulatory feedbacks, and their spatio-temporal deployment, the interpretation of the outcome for a particular regulatory interaction may not be intuitive. To overcome this limitation, several studies have used an iterative approach combining experimental quantifications and computational modelling that allowed to predict and reveal new regulatory nodes mediating SAM activity (*Besnard et al., 2014*; *Chickarmane et al., 2012*).

From a dynamic point of view, cells produced in the central stem cell domain are continuously displaced towards the periphery by the divisions of cells located more centrally until they are incorporated into an organ primordium and finally differentiate. Hence, they undergo two major fate transitions: First, the transition from a slowly dividing stem cell to a more rapidly dividing transit amplifying cell and second, the transition from a non-committed amplifying cell to primordium cell by passing the peripheral zone. Despite the identification of key regulators controlling shoot stem cell activity, and patterning different domains of the SAM, our current understanding of how cell fate progression is modulated remains largely elusive. We used an integrated approach combining live-cell imaging, computational modelling and genome-wide profiling to show that *HEC* function modulates the stepwise fate transition from CZ cell to PZ and from PZ to lateral organ cell fate by coordinating the balance between cytokinin and auxin responses.

## Results

### HECATE genes regulate cell fate transition at the SAM

In order to dynamically trace cell-behaviour at the SAM, we established an image analysis pipeline that allows identification and quantification of cells within specific sub-domains of the SAM. When we applied this tool to plants that lack *HEC* function (*hec1,2,3* triple mutants), we found that their SAMs were smaller and displayed a reduced cell number, in line with our previous results (*Figure 1A–D*) (*Schuster et al., 2014*). Average cell size was not affected in SAMs of *hec1,2,3* triple mutants, supporting that reduction in meristem size results from decreased cell number (*Figure 1E–*

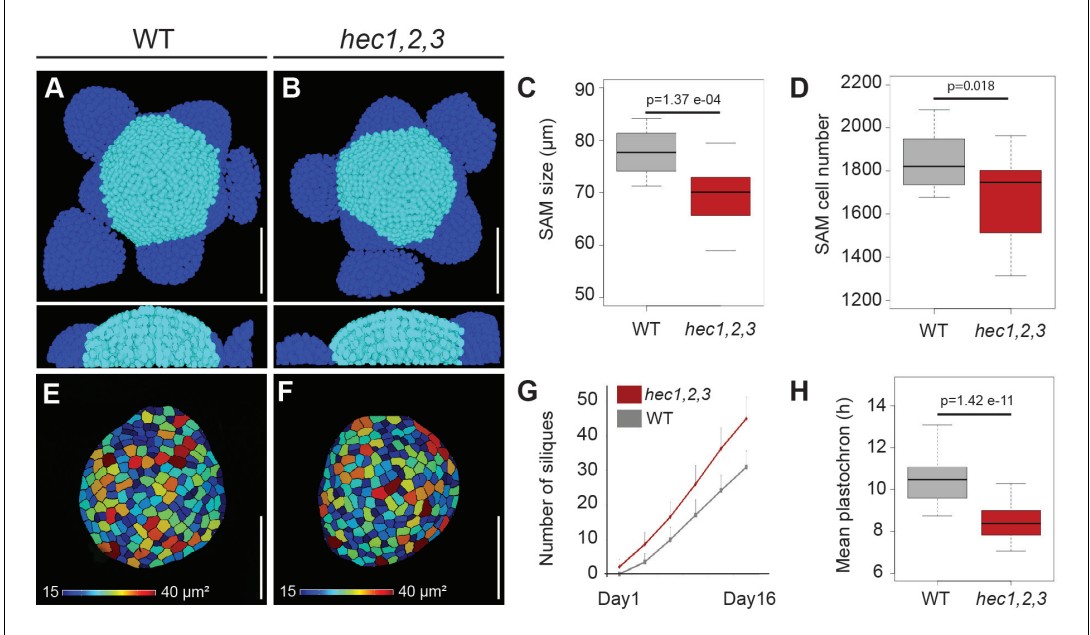

**Figure 1.** SAM size and organ initiation rate are uncoupled in *hec1,2,3*. (A–B) Representative views of 3D-reconstructed shoot meristems after nuclei segmentation from WT (A) and *hec1,2,3* (B). Light blue: SAM cells; dark blue: primordia cells. (C) Shoot apical meristem size at 28 days after germination (DAG) (n = 15) (D) Quantification of SAM cell number in WT (n = 19) and *hec1,2,3* (n = 21). (E–F) Representative cell area of segmented L1 layer from WT (E) and *hec1,2,3* (F) SAM (n > 3) (G) Cumulated number of siliques over time in WT (n = 46) and *hec1,2,3* (n = 42). (H) Mean inflorescence plastochron in WT (n = 46) and *hec1,2,3* (n = 42). Scale bars: 50 µm. Statistical test: Student t- test (C,H), Wilcoxon signed-ranked test (D).
DOI: https://doi.org/10.7554/eLife.30135.003

The following source data is available for figure 1:

**Source data 1.** SAM size quantification (panel C);
DOI: https://doi.org/10.7554/eLife.30135.004

*F*). Surprisingly, the average time interval between the initiation of successive lateral organs along the stem (plastochron) was substantially reduced in *hec1,2,3* plants, despite the smaller SAM (*Figure 1G–H*). The conflicting phenotype of a smaller stem cell system that produced more organs per time led us to hypothesise that rather than exclusively acting on stem cells (*Schuster et al., 2014*), *HEC* genes may have a broader function to coordinate the acquisition of cellular identity and thus cell behaviour across different regions of the SAM.

Since the regulatory network underlying SAM activity is strongly stabilized by feedbacks, whereas cell fate transitions are inherently dynamic, we decided to study the function of *HEC* genes by time-resolved live-imaging. To this end, we created lines, which allowed us to experimentally control *HEC* activity in space and time by fusing the coding sequence of *HEC1* to the glucocorticoid receptor (GR) domain from rat via a flexible linker. The resulting HEC1-linker-GR protein remained trapped in the cytoplasm and its activity could be induced by treatment with the steroid dexamethasone (dex). Given the limited capacity of HEC-linker-GFP for lateral cell-to-cell movement in the SAM (*Figure 2— figure supplement 1*; *Daum et al., 2014*), inducing the even larger HEC1-linker-GR protein allowed us to assess HEC function locally in different domains of the meristem.

First, we analysed *HEC1* function in the CZ using lines carrying *pCLV3:HEC1-linker-GR* and imaged plants daily after dex or mock treatment. In contrast to the mock control, dex treatment caused a gradual enlargement of the SAM over time, which was in line with our observation of constitutive *pCLV3:HEC1-linker-GFP* lines (*Figure 2—figure supplement 2A,B*). To investigate the regulatory mechanisms underlying the observed increase in SAM size, we quantified expression intensities and domain sizes of the key stem cell regulators *WUSCHEL* (*WUS*) and *CLAVATA3* (*CLV3*) by fluorescent markers, which faithfully label niche (*pWUS:3xYFP-NLS*) and stem cells (*pCLV3: mCherry-NLS*). Interestingly, the increase in SAM size caused by stem cell-specific activation of *HEC1* was accompanied by a transient enlargement of both the niche, or organising centre (OC),

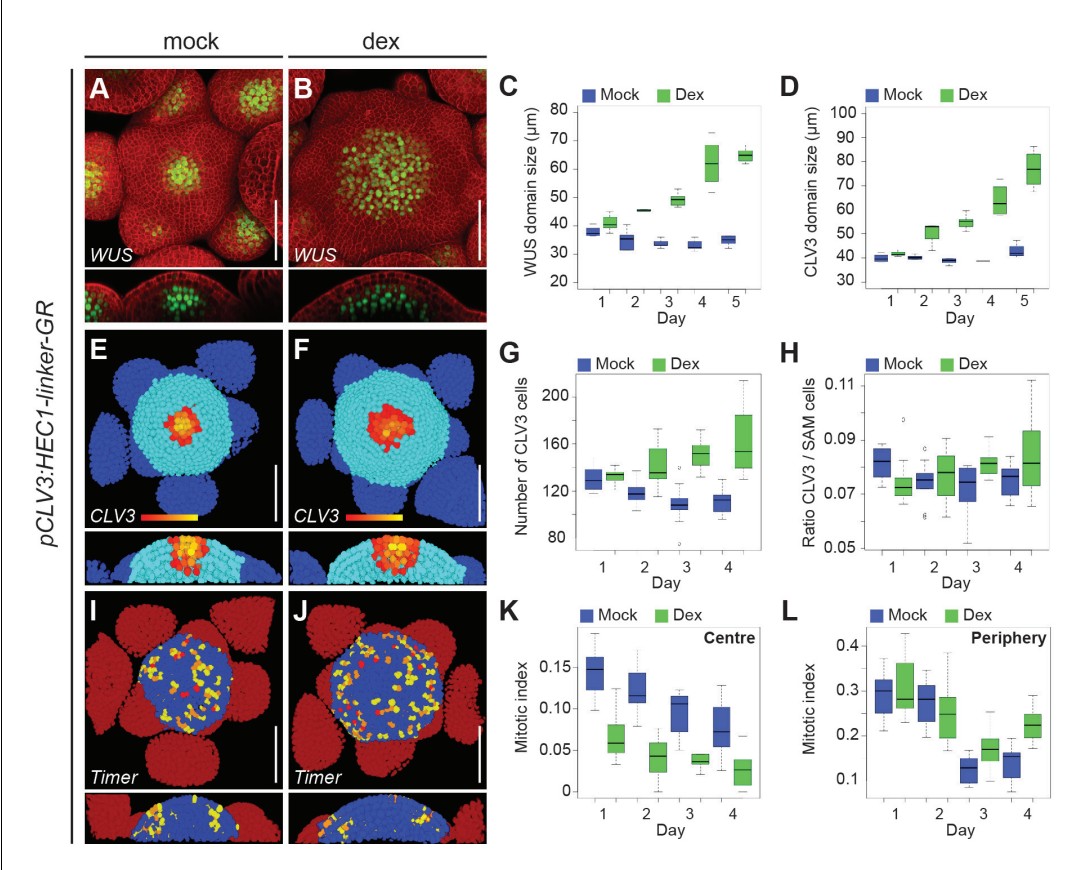

**Figure 2.** *HEC* function controls CZ to PZ fate transition. (**A–B**) Representative expression of *pWUS:3xYFP-NLS* in SAMs of *pCLV3:HEC1-linker-GR* plants four days after mock (**A**) or dex (**B**) treatment. (**C–D**) Development of *WUS* (**C**) and *CLV3* (**D**) domain sizes in *pCLV3:HEC1-linker-GR* SAMs after mock or dex treatment. (**E–H**) Analysis *of pCLV3:mCherry-NLS* expression after image segmentation. Representative images of segmented SAMs four days after mock (**E**) or dex (**F**) treatment of *pCLV3:HEC1-linker-GR* plants. Yellow, orange, red highlight *CLV3* positive cells. Quantification of *CLV3* positive cell number (**G**) and ratio between *CLV3* and total SAM cell number (**H**) after mock or dex treatment (n > 9 per condition). (**I–J**) Cell proliferation (*pKNOLLE: mFluorescentTimer-NLS*) in *pCLV3:HEC1-linker-GR* plants two days after mock (**I**) or dex (**J**) treatment. Red, orange, yellow: recently divided cells, blue: older cells; dark red: primordia (n > 7 per condition). (**K–L**) Time series quantification of mitotic index (young dividing cells/total cell number) at the centre (**K**) or at the periphery (**L**) of the SAM in *pCLV3:HEC1-linker-GR/pKNOLLE:fast-mFluorescentTimer-NLS* after mock or dex treatment. Scale bar: 50 μm.

DOI: https://doi.org/10.7554/eLife.30135.005

The following source data and figure supplements are available for figure 2:

**Source data 1.** Intensity plot profiles: *pWUS:3xYFP-NLS* (panel C).
DOI: https://doi.org/10.7554/eLife.30135.012

**Source data 2.** SAM size measurement *pCLV3:HEC1-linker-GR* (*Figure 2—figure supplement 2A*);
DOI: https://doi.org/10.7554/eLife.30135.013

**Figure supplement 1.** HEC1-linker-GFP protein mobility in the SAM.
DOI: https://doi.org/10.7554/eLife.30135.006

**Figure supplement 2.** SAM behavior after promoting of HEC function at the CZ.
DOI: https://doi.org/10.7554/eLife.30135.007

**Figure supplement 3.** Dynamics of *WUS* and *CLV3* reporters after HEC1-GR induction at the CZ.
DOI: https://doi.org/10.7554/eLife.30135.008

**Figure supplement 4.** Intensity plot profiles of *WUS* and *CLV3* reporters after HEC1-GR induction at the CZ.
DOI: https://doi.org/10.7554/eLife.30135.009

**Figure supplement 5.** Time series of 3D-reconstructed *pCLV3:HEC1-linker-GR* shoot meristems expressing *pCLV3:mCherry-NLS/UBQ10:3xGFP-NLS* after mock or dex induction.
DOI: https://doi.org/10.7554/eLife.30135.010

**Figure supplement 6.** Dynamics of the SAM proliferative activity after HEC1-GR induction at the CZ.
DOI: https://doi.org/10.7554/eLife.30135.011

and the CZ (*Figure 2A–G*; *Figure 2—figure supplements 3–5*). After an initial expansion of expression domains, we found that the intensities of *WUS* and *CLV3* reporters were strongly reduced at day 4, likely as an effect of a feedback mechanism restricting CZ and OC fate (*Figure 2—figure supplement 4*). This indirect negative effect of *HEC1* on *WUS* and *CLV3* expression explained our previous findings using stably expressing transgenes, which had shown a reduction in *WUS* and *CLV3* expression (*Schuster et al., 2014*) and revealed that *HEC1* promotes SAM fasciation via transient stimulation of *WUS* (*Figure 2A–C*, *Figure 2—figure supplements 3* and *4B*).

The observed short-term increase in stem cell number could result from different *HEC1*-dependent mechanisms: (1) a local increase of stem cell proliferation; (2) re-specification of early PZ cells into stem cells; and (3) a reduction in the differentiation rate between stem cells and the PZ. To discriminate between these scenarios, we developed a novel imaging tool combining a fluorescent timer protein (*Subach et al., 2009*) driven from a cell cycle dependent promoter (*pKNOLLE:fast-mFluorescentTimer-NLS*) with an ubiquitously and homogeneously expressed GFP (*pUBQ10:3xGFP-NLS*). The timer protein was exclusively expressed during cytokinesis and then slowly matured from a blue form to a form that exhibits red fluorescence. Therefore, the ratio of the fading blue to stable green signal could be used as a readout for time passed since the last division, allowing us to assess the age of cells and thus served as a proxy for division frequency. Interestingly, we observed that although the SAM expanded, cell division activity after stem cell specific induction of *HEC1* was decreased at the centre of the SAM and mostly confined to the PZ as in mock-treated plants (*Figure 2I–L*; *Figure 2—figure supplement 6*; Materials and methods section). This demonstrated

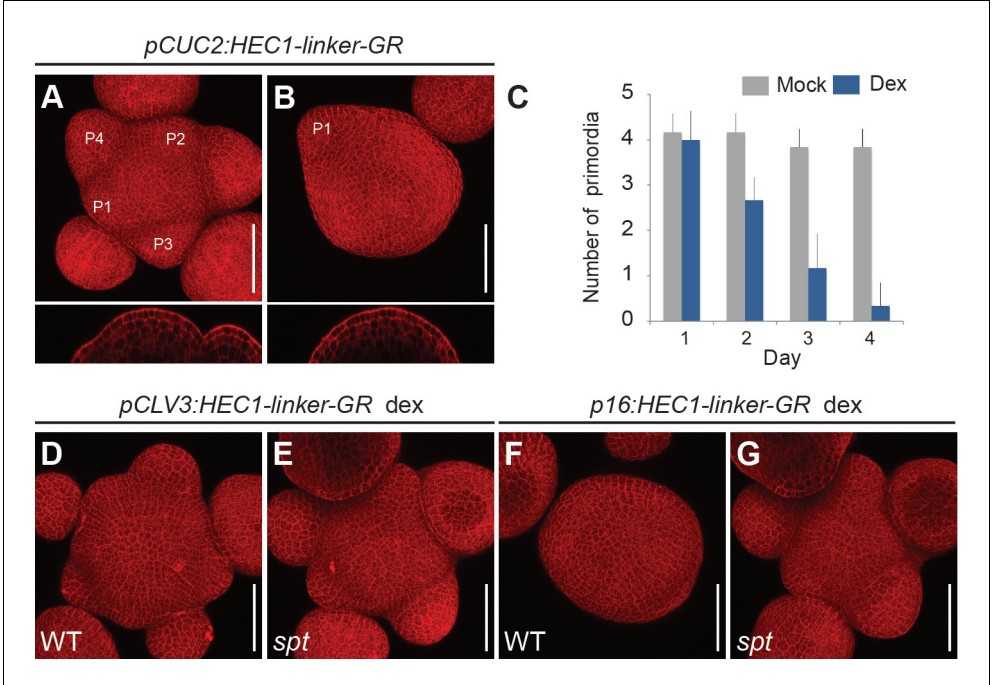

**Figure 3.** *HEC* function controls PZ to organ primordia fate transition. (**A–B**) Representative images of *pCUC2:HEC1-linker-GR* four days after mock (**A**) or dex treatment (**B**). (**C**) Quantification of primordia number formed in *pCUC2:HEC1-linker-GR* after mock and dex treatment over time (n = 6 per condition). (**D–E**) Representative view of *pCLV3:HEC1-linker-GR* four days after dex treatment in WT (**D**) or *spt* mutant background (**E**) (WT: n = 5; *spt*: n = 9). (**F–G**) Representative view of *p16:HEC1-linker-GR* four days after dex treatment in WT (**F**) or *spt* background (**G**) (WT: n = 6; *spt*: n = 10). Scale bar: 50 μm.

DOI: https://doi.org/10.7554/eLife.30135.014

The following source data and figure supplement are available for figure 3:

**Source data 1.** Quantification primordia number (panel C).
DOI: https://doi.org/10.7554/eLife.30135.016
**Figure supplement 1.** SAM behaviour after promoting HEC function at the PZ/BZ.
DOI: https://doi.org/10.7554/eLife.30135.015

that *HEC1* did not locally promote stem cell division. In contrast, we observed an increase in the mitotic index of the PZ at later stages (day 3 and day 4) despite the fact that *HEC1* protein is largely unable to move from cell to cell, suggesting that cell proliferation in the periphery was stimulated non-cell autonomously (*Figure 2L*; *Figure 2—figure supplement 6C*). Next, we addressed fate respecification of early PZ cells into stem cells. To this end, we analysed the ratio between *CLV3* positive cells and all SAM cells after stem cell specific expression of *HEC1* (*Figure 2E–H*, *Figure 2—figure supplement 5*). Interestingly, although the number of *CLV3* positive cells strongly increased after dex treatment, the ratio between *CLV3* and all SAM cells remained essentially unchanged (*Figure 2G–H*). This argued against reprogramming of early PZ progenitor to stem cells by *HEC1*, but rather supported the idea of coordinated cell behaviour in CZ and PZ (*Figure 2—figure supplements 3–5*). Taken together, these two lines of experiments excluded that cell proliferation in the CZ or re-specification of early PZ cells were the main drivers of *HEC1*-mediated meristem expansion and left us with a model in which *HEC1* activity would locally inhibit the transition from stem cell to PZ fate, giving rise to a larger stem cell domain and consequently to an enlarged shoot meristem.

This idea was in line with the *hec1,2,3* mutant phenotype and suggested that in these plants, stem cells would transit more quickly from stem cell to PZ fate and further on to organ fate. To test this model, we next asked whether *HEC1* could also interfere with the PZ to organ transition. Therefore, we increased *HEC1* activity at the periphery and boundary zone (BZ) either by stable *pCUC2:HEC1-linker-GFP* or transient *pCUC2:HEC1-linker-GR* expression (*Figure 3A–B*; *Figure 3—figure supplement 1*). Strikingly, *HEC1* induction at the periphery and BZ gradually supressed the emergence of lateral organs, eventually resulting in the formation of pin-like inflorescences, demonstrating that *HEC1* can potently interfere with incorporation of cells into organs when expressed at the BZ (*Figure 3C*, *Figure 3—figure supplement 1G*).

To test the biological relevance of these results, we combined domain specific activation of HEC activity with constitutive loss of function of the interacting partner SPT, which we had shown to be required for *HEC1* output in stem cells and flowers (*Gremski et al., 2007*; *Schuster et al., 2014*). Strikingly, the developmental phenotypes observed in distinct domains of the SAM were fully suppressed in the *spt* mutant background, including the formation of pin-like inflorescences after activation of HEC1 at the periphery (*Figure 3D–G*). These results underlined a relevant function of the HEC-SPT protein complex in controlling SAM dynamics (*Figure 3D–G*).

Taken together, *HEC* function appeared not only to control stem cell to PZ, but also to be required and sufficient for PZ to organ fate transitions together with its partner SPT and thus acted as a central gatekeeper for cell fate progression throughout the SAM.

## HECATE function controls the dynamics of cell differentiation

Our results from live-cell imaging had shown that *HEC* genes were able to control CZ to PZ as well as PZ to organ primordia cell fate transitions, and promote cell proliferation at the SAM periphery. While consistent with our observation of *hec1,2,3* mutants having smaller meristems that initiate more lateral organs, these findings were insufficient to formally explain the changes in cell behaviour underlying *HEC* loss-of-function. Due to the static nature of the triple mutant, which precluded time resolved experimental analysis, we developed a computational model to elucidate the role of *HEC* factors in regulating the dynamics of cell fate progression at the SAM by simulations.

First, we established and calibrated a cell population model by defining model parameters based on published data and quantitative in vivo imaging results of stem cell, peripheral cell and organ numbers (*Figure 4A*). We specified primordia initiation rate (2.3 per day; [*Figure 1H*]), CZ and PZ proliferation rate (39.8 and 18.3 hr respectively; [*Reddy et al., 2004*]) and primordia separation time after their initiation (2.2 days; [*Besnard et al., 2014*]). Using these empirically derived parameters, the calibrated model was robust to perturbations and converged to a unique dynamic state of balanced cell proliferation, organ formation and separation (*Figure 4B*; *Figure 4—figure supplement 1A*).

We then simulated the *HEC* loss-of-function scenario by increasing the differentiation rate between the CZ, PZ and primordia and compared the resulting dynamics with time-resolved data obtained from in vivo SAM imaging. Although the resulting CZ and total SAM cells number fitted our experimental measurements, the cumulated number of primordia did not increase as observed in *hec1,2,3* mutant plants (*Figure 4C*). Thus, we further tested the impact of modulating the primordia initiation rate on SAM cell behaviour. By combining an increased CZ to PZ transition with an

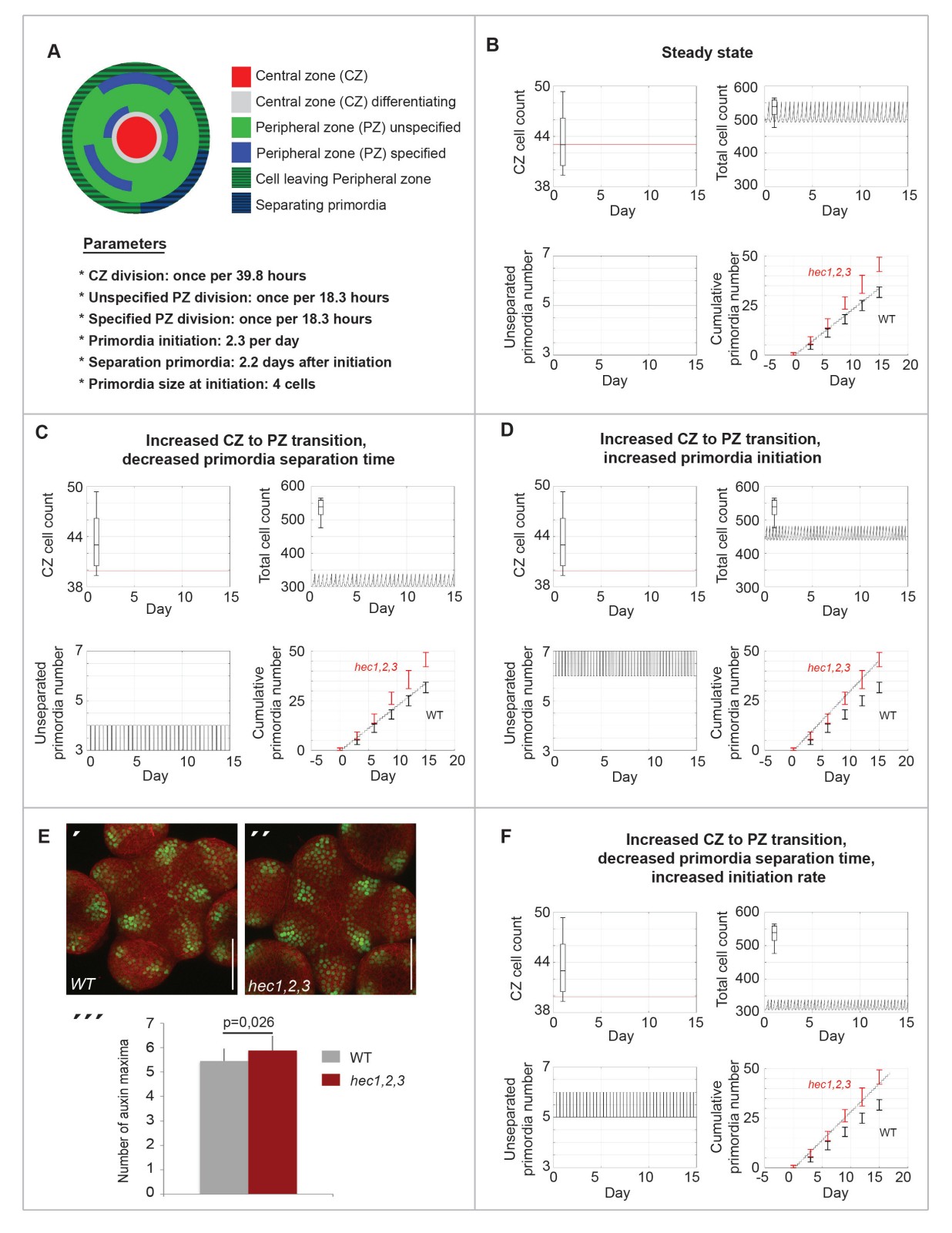

**Figure 4.** *HEC* genes control cell differentiation dynamics in the SAM. Computational simulations displaying in silico estimation (line) and observed in vivo quantification (boxes) for CZ cell number, total cell number, CZ/total cell ratio, number of unseparated primordia and cumulative number of primordia. (**A**) Description of computational model. (**B**) Calibration of SAM model. (**C–D**) Simulation of HEC loss-of function on SAM cell behaviour. (**C**) Effects caused by increasing cell differentiation between CZ and PZ and decreasing primordia separation time. (**D**) Simulation of effects caused by

*Figure 4 continued on next page*

*Figure 4 continued*

increasing CZ to PZ transition and increasing primordia initiation rate. (**E**) Analysis of *pDR5v2:3xYFP-NLS* in WT (´) and *hec1,2,3* (´´) SAMs. (´´´)
Quantification of auxin maxima in WT (n = 20) and *hec1,2,3* (n = 17). (**F**) Simulation of effects caused by increasing CZ to PZ transition, increasing
primordia initiation rate and decreasing their separation time. Cell numbers in (**B,C,D,F**) refer to a single cellular layer and correspond to one third of
the respective quantified cell numbers. Scale bar: 50 µm (**E**). Statistical test: Student t-test (E´´´).

DOI: https://doi.org/10.7554/eLife.30135.017

The following source data and figure supplement are available for figure 4:

**Source data 1.** Quantification auxin maxima pDR5v2:3xYFP-NLS (panel E).
DOI: https://doi.org/10.7554/eLife.30135.019
**Figure supplement 1.** Computational model simulations.
DOI: https://doi.org/10.7554/eLife.30135.018

increased primordia initiation rate, our simulations could reproduce experimental measurements for the number of CZ cells, SAM cells and for the cumulated number of lateral organs (*Figure 4D*). Furthermore, these model simulations predicted that increasing initiation rate would lead to a larger number of unseparated organ primordia at any given point in time, which could experimentally be tested by assessing the number of auxin output maxima as a proxy. Therefore, we introduced the auxin output reporter *pDR5v2:3xYFP-NLS* (*Liao et al., 2015*) into the *hec1,2,3* triple mutant and quantified the number of auxin output maxima (*Figure 4E*). However, in contrast to the model prediction, we did not observe a major increase in the number of DR5 positive domains in SAMs of *hec1,2,3* compared to wild type (5.25 in wild type; 5.70 in *hec1,2,3*; n > 16)(*Figure 4E´´´*). While the difference was statistically significant, it was substantially smaller than predicted by the model and insufficient to explain the increase in organ formation (*Figure 4D*). Hence, we needed to test additional factors in our model for their contribution to the *hec* loss-of-function phenotype. Having experimentally assessed the contribution of organ initiation rate in *hec1,2,3* now allowed us to fix this important parameter for wild type and mutant plants and to further explore the parameter space for the time organs take to separate from the SAM after initiation. Indeed, reducing the time from initiation to separation of organs from 52 hr in wild type to 42 hr in *hec1,2,3* combined with a slight increase in the rate of organ initiation in the mutant resulted in simulations fitting all our experimental results (*Figure 4F*). Importantly, our simulations not only qualitatively captured the dynamics of *HEC* loss-of-function meristem but also allowed us to compute that in *hec1,2,3*, primordia were initiated at a 15% higher rate and separated 10 hr earlier from the SAM than in wild type, supporting the idea that *HEC* function modulates the dynamics of stem cell differentiation.

To further test our model, we next simulated *HEC1* gain-of-function experiments in the CZ (*Figure 4—figure supplement 1B–E*). In line with our hypothesis and with the results of the quantified mitotic index, combining a reduction in stem cell to peripheral fate transition with an increase in peripheral cell proliferation was sufficient to recapitulate our experimental observations (*Figure 4—figure supplement 1F*). On the other hand, delaying CZ to PZ transition only, increasing the proliferation rate in the PZ only or introducing a re-specification of PZ cells into CZ cells did not reproduce our in vivo data (*Figure 4—figure supplement 1B–E*).

Taken together, applying reiterative cycles of experimentation and modelling allowed us to derive a quantitative framework of *HEC* function in the SAM. Simulating loss- and gain-of-function experiments, our model faithfully captured the dynamics of the SAM and quantitatively supported our hypothesis that HEC factors modulate the rate of cell differentiation at the shoot meristem at multiple levels.

## *HECATE* genes locally modulate auxin and cytokinin signalling

After having shown that *HEC* genes control SAM cell behaviour, we wondered what the underlying molecular mechanisms might be. The observation that HECs locally inhibited cell fate progression and non-cell autonomously stimulated proliferation suggested that they could modulate cell-to-cell communication. Thus, we first analysed the response of the auxin and cytokinin systems, two key phytohormones controlling cell fate and cell proliferation at the SAM (reviewed in [*Gaillochet et al., 2015*]).

To investigate the involvement of cytokinin, we created gain- and loss-of-HEC-function plants that carried the *pTCSn:erGFP* cytokinin output reporter (*Zürcher et al., 2013*). In contrast to WT,

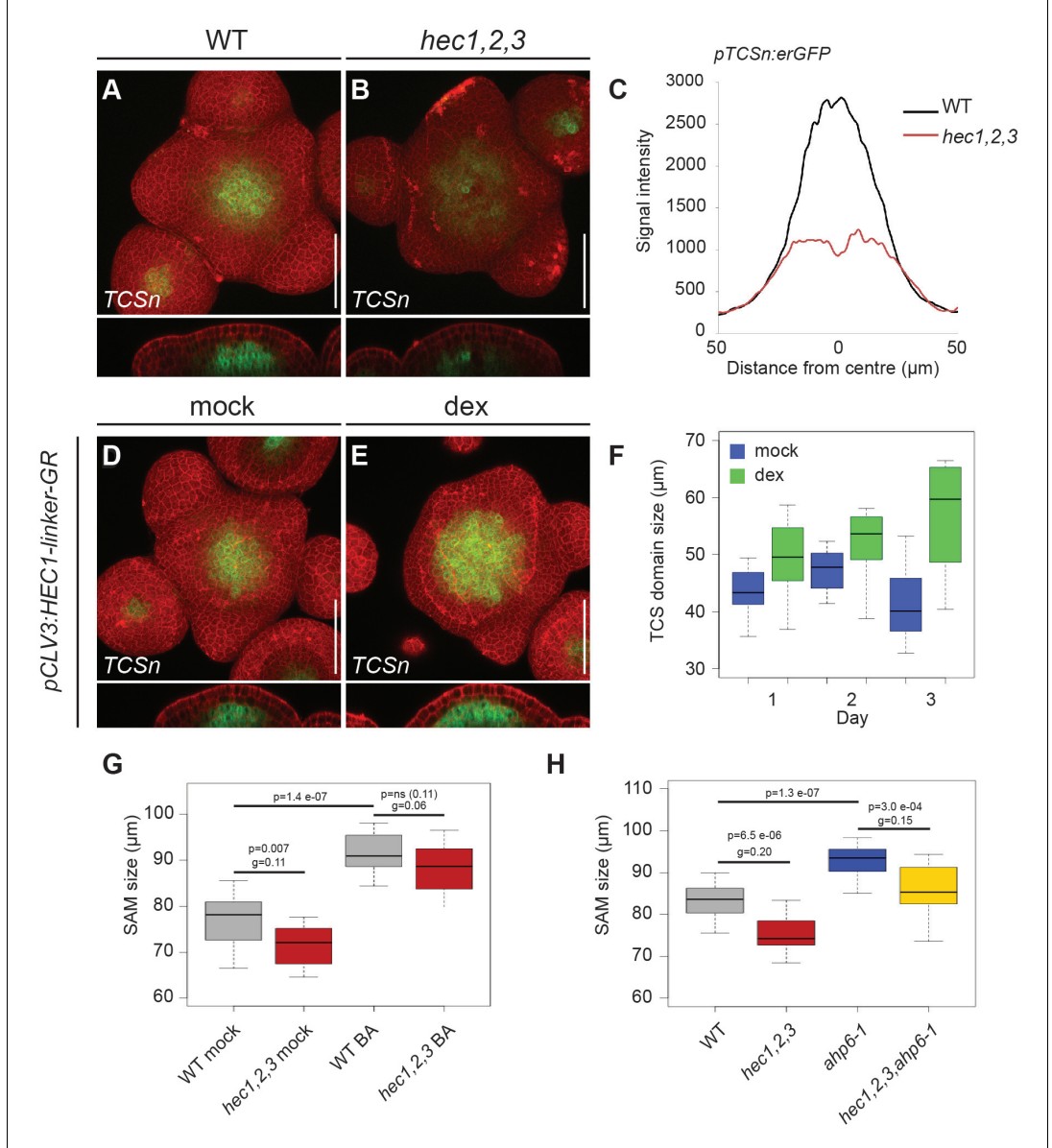

**Figure 5.** *HEC* genes promote cytokinin signalling. (A–B) Representative expression of *pTCSn:erGFP* in WT (**A**) and *hec1,2,3* (**B**) SAMs. (**C**) Average intensity plot profile of *pTCSn:erGFP* signal in WT and *hec1,2,3* SAM (n > 14 per genotype). (D–F) Analysis of *pTCSn:erGFP* activity in response to *pCLV3:HEC1-linker-GR* induction. (D–E) Representative views of SAMs three days after mock (**D**) or dex treatment (**E**) (n > 6 per condition). (**F**) Development of TCS domain size after mock (blue) or dex (green) treatment. (**G**) Shoot meristem size after cytokinin treatment in WT (n > 13) and *hec1,2,3* (n > 13) (**H**) Shoot meristem size after genetically modulating cytokinin signalling in WT (n > 14) and *hec1,2,3* (n > 14). Statistical test: Student t-test; Effect size: Hedges´coefficient g (**G,H**). Scale bar: 50 µm.

DOI: https://doi.org/10.7554/eLife.30135.020

The following source data and figure supplement are available for figure 5:

**Source data 1.** Intensity plot profiles *pTCSn:erGFP* (panel C);
DOI: https://doi.org/10.7554/eLife.30135.022
**Figure supplement 1.** Dynamics of cytokinin responses after modulation of HEC function.
DOI: https://doi.org/10.7554/eLife.30135.021

*hec1,2,3* triple mutants displayed a substantial reduction in *pTCSn:erGFP* signal specific to the SAM, whereas TCS activity in the root was unchanged (***Figure 5A–C***; ***Figure 5—figure supplement 1A–B***). Conversely, increased *HEC1* activity in stem cells led to a significant expansion of the central

cytokinin-signalling domain, which was concomitant with SAM enlargement (*Figure 5D–F*; *Figure 5—figure supplement 1C*). Induction of *HEC1* at the PZ and BZ did not locally promote cytokinin signalling, highlighting the domain-specific activity of HEC factors (*Figure 5—figure supplement 1D–E*).

To further investigate the functional interaction between HECs and cytokinin, we tested whether promoting cytokinin signalling was sufficient to rescue the reduction in SAM size observed in *hec1,2,3* plants. Therefore, we enhanced cytokinin signalling either chemically or genetically by 6-Benzylaminopurine (BA) treatment or removal of AHP6, a negative component of cytokinin signal transduction (*Besnard et al., 2014*; *Mähönen et al., 2006*), respectively. In our growth conditions, both chemical and genetic stimulation largely suppressed SAM size defects of *hec1,2,3* SAMs (*Figure 5G–H*). Permanent inactivation of *AHP6* by the *ahp6-1* mutation caused SAM expansion in both wt and *hec1,2,3*, but the mutants responded more strongly, as shown by the reduced SAM size difference between WT and *hec1,2,3* (Hedges´ g coefficient decreased from 0.20 to 0.15) (*Figure 5H*). Treatment with 50 µM BA for 8 days showed a similar trend with SAM expansion in both genotypes and a more pronounced response in *hec1,2,3* (Hedges´ g coefficient decreased from 0.11 to 0.06)(*Figure 5G*). While after mock treatment mutants exhibited SAMs of 93% the wt size, BA treatment almost fully supressed this phenotype and *hec1,2,3* SAMs were now increased to 97%, which was not significantly different from treated wt apices (*Figure 5G*).

Together, these results showed that *HEC* function was sufficient and required to promote cytokinin signalling, which subsequently affected SAM size. This also suggested a mechanism for the expansion of the OC and CZ after stem cell specific induction of *HEC1*: A non-cell autonomous stimulation of cytokinin signalling by *HEC1* could trigger the activation of *WUS*, which in turn would promote stem cell fate.

To analyse the interplay between HEC activity and auxin signalling, we next monitored auxin sensing and downstream transcriptional output using the *R2D2* and *pDR5v2:3xYFP-NLS* reporters (*Liao et al., 2015*), respectively, in *HEC* gain- and loss-of-function plants. Our analysis showed that the topology of auxin signalling input and output in the SAM was only mildly changed in *hec1,2,3* compared to WT (*Figure 4E*; *Figure 6—figure supplement 1*), suggesting that auxin signalling does not critically depend on *HEC* function. However, the observation of a small but significant increase in the number of DR5v2 positive auxin maxima (*Figure 4E´´´*), led us to hypothesise that HEC factors might impinge on the auxin feedback system and thus may quantitatively modulate signalling output. To test this hypothesis, we recorded auxin responses after induction of *HEC1* at the SAM periphery using *pCUC2:HEC1-linker-GR*. Consistent with the increase in auxin output observed in *hec1,2,3* mutants, boosting *HEC* activity at the periphery led to a substantial reduction in the number of DR5 signal maxima, eventually bringing about a complete collapse of lateral organ initiation (*Figure 6A–C*; *Figure 6—figure supplement 2A–C*). Importantly, inducing *HEC1* in stem cells repressed auxin perception locally but did not change auxin responses at the site of primordia initiation, demonstrating that *HEC1* controls auxin signalling strictly cell autonomously (*Figure 6—figure supplement 2D–G*). Taken together, these results showed that *HEC* function regulates cellular sensitivity to cytokinin and auxin signals in a domain-specific manner.

In addition to their critical role in the SAM, auxin and cytokinin determine cell fate acquisition in the root apical meristem (RAM). However, in contrast to the SAM, auxin promotes stem cell fate and cytokinin signalling marks the entry into differentiation (reviewed in [*Gaillochet and Lohmann, 2015*]). Since *HEC* genes are only very weakly expressed in the RAM (*Figure 6—figure supplement 3A*) (*Li et al., 2016*), the root is ideally suited to test the ability of *HEC1* to control cell fate transition through the modulation of auxin and cytokinin signalling independent of SAM specific feedback systems. To this end, we generated lines that combined inducible expression of *HEC1* in proliferative tissues (*Figure 6—figure supplement 3B*), including the root tip (*p16:HEC1-linker-GR*), and the output reporters for cytokinin and auxin, *pTCSn:erGFP* and *pDR5v2:3xYFP-NLS*, respectively (*Figure 6—figure supplement 3C–R*). Strikingly, we observed a reduction of RAM size and meristem cell number after induction of *p16:HEC1-linker-GR*, while cell division activity and cell size were unaffected (*Figure 6—figure supplement 3C–L*). These developmental changes in the RAM correlated well with increased cytokinin and decreased auxin signalling at the transition zone (*Figure 6—figure supplement 3M–P*). Furthermore, the elongation zone was substantially reduced as marked by the development of root hairs and we frequently observed ectopic periclinal divisions within the cortical layer two days after induction (*Figure 6—figure supplement 3F and Q–R*). Taken together, these

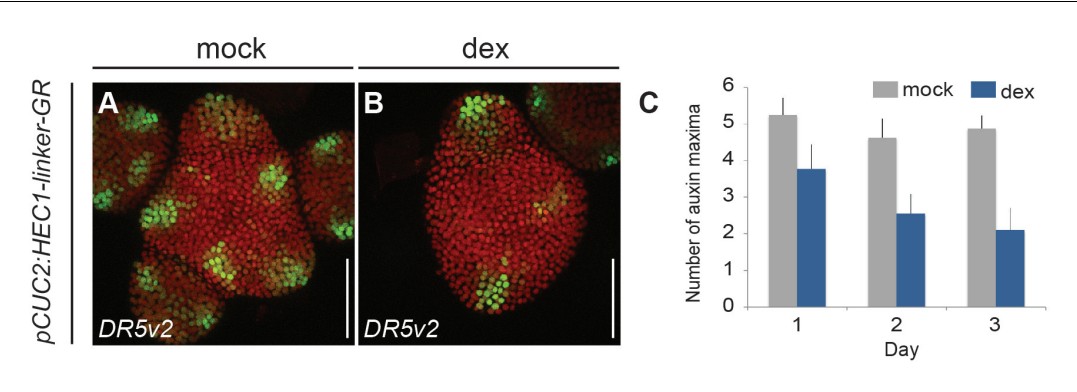

**Figure 6.** *HEC* function modulates auxin signalling. (**A–B**) *pDR5v2:3xYFP-NLS* expression three days after mock (**A**) or dex treatment (**B**) of *pCUC2: HEC1-linker-GR/pRPS5a:mCherry-NLS* plants. (**C**) Quantification of auxin maxima in *pCUC2:HEC1-linker-GR* after mock or dex treatment (n > 7 per condition) over time. Scale bar: 50 µm.

DOI: https://doi.org/10.7554/eLife.30135.023

The following source data and figure supplements are available for figure 6:

**Source data 1.** Quantification auxin maxima SAM (panel C).

DOI: https://doi.org/10.7554/eLife.30135.027

**Source data 2.** Quantification number of auxin maxima (*Figure 6—figure supplement 2C*);

DOI: https://doi.org/10.7554/eLife.30135.028

**Figure supplement 1.** Auxin responses at the SAM in *HEC* loss-of-function mutants.

DOI: https://doi.org/10.7554/eLife.30135.024

**Figure supplement 2.** Dynamics of auxin responses at the SAM in *HEC* gain-of-function mutants.

DOI: https://doi.org/10.7554/eLife.30135.025

**Figure supplement 3.** RAM developmental dynamics in *HEC* gain-of-function mutants.

DOI: https://doi.org/10.7554/eLife.30135.026

results demonstrated that *HEC* activity was sufficient to modulate phytohormonal balance in diverse cellular contexts independent of the regulatory environment and underlined its central role in regulating the crosstalk between auxin and cytokinin responses.

## Molecular network underlying HEC function

After having shown that *HEC1* likely works via modulation of auxin and cytokinin pathways at the SAM, we next aimed at dissecting the transcriptional regulatory network orchestrated by HEC factors, using HEC1 as a proxy. To this end, we used genome-wide profiling to identify early HEC1 response genes (*Figure 7—figure supplement 1A*; *Supplementary file 1*). First, we recorded HEC1 DNA binding pattern using ChIP-seq on a functional *p35S:HEC1-linker-GFP* line. We found 6930 binding regions of HEC1 in 5250 unique genes with 74.5% of the events located within 3 kb upstream of transcriptional start sites (*Figure 7A*). The HEC1 DNA binding pattern was distinct from those of other bHLH transcription factors, suggesting that our ChIP-seq had indeed captured the chromatin binding universe of HEC1 (*Figure 7—figure supplement 1C*). To complement the binding data, we recorded inflorescence-specific HEC1 response genes by RNA-seq analysis using micro-dissected shoot apices of our *p16:HEC1-linker-GR* line. We identified 957 significantly regulated genes after three hours of HEC1 induction by dex and 815 transcripts after induction by dex and simultaneous inhibition of protein biosynthesis by cycloheximide (cyc) (p<0.05) (*Figure 7—figure supplement 1A*; *Supplementary file 1*). We were able to confirm the direct regulation of *PIN3*, which we had previously shown, suggesting that our experimental strategy was successful (*Figure 7—figure supplement 1B*) (*Schuster et al., 2015*). Surprisingly, we only found a few canonical components of the auxin or cytokinin signalling circuitries among the direct targets, suggesting that HEC regulators do not have switch-like properties for these pathways (Table 6 in *Supplementary file 1*). However, we found a significant overlap between HEC1-response genes and genes responsive to cytokinin, suggesting that these two regulatory pathways also converge at the molecular level (*Figure 5*; *Figure 7—figure supplement 1D*).

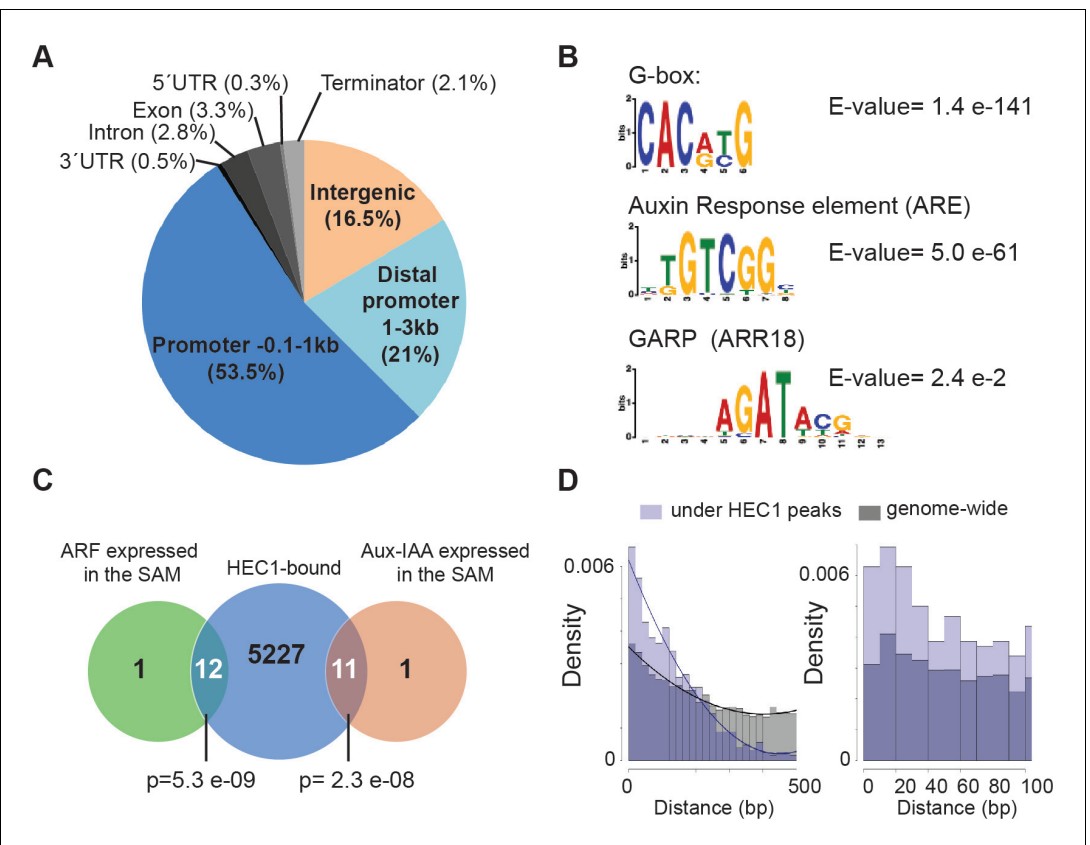

**Figure 7.** *HEC*1 DNA binding profile. (**A**) Genome-wide distribution of HEC1-bound regions relative to gene models. (**B**) Motif enrichment analysis from HEC1 ChIP-seq. Position weight matrix depicting G-box, ARE and GARP motifs and corresponding E-value. (**C**) Venn diagram showing overlap between HEC1-bound genes and ARF and Aux-IAAs expressed in the SAM. (**D**) Distribution of distances between G-box and ARE under HEC1 peaks (light purple) and on the genomic background (grey). Overlap: dark purple. Fitting curves correspond to polynomial $2^{nd}$ order fit. Statistical test: hypergeometric test (**C**).

DOI: https://doi.org/10.7554/eLife.30135.029

The following source data and figure supplements are available for figure 7:

**Source data 1.** Calculation ARE_G-box enrichments (*Figure 7—figure supplement 1G–I*).
DOI: https://doi.org/10.7554/eLife.30135.032
**Figure supplement 1.** Analysis of HEC regulatory patterns in genome-wide profiles.
DOI: https://doi.org/10.7554/eLife.30135.030
**Figure supplement 2.** Visualization of HEC1 binding peaks at the loci of ARFs and Aux-IAAs expressed in the SAM.
DOI: https://doi.org/10.7554/eLife.30135.031

Next, we carefully analysed the identified binding regions and in line with the quality of our data-set, one of the most highly enriched DNA motif in HEC1 binding regions was a G-Box, the sequence known to be the preferentially bound by bHLH transcription factors (E-value = 1.4 e-141) (*Figure 7B*) (*Lau et al., 2014*; *Pfeiffer et al., 2014*). Interestingly, auxin response elements (ARE), the DNA cis-regulatory motifs targeted by ARF transcription factors to regulate auxin dependent gene expression, were also significantly over-represented under HEC1 peaks (ARE, E-value = 5.0 e-61). In contrast, GARP elements, bound by type B-ARRs, the cytokinin output transcription factors, were only mildly enriched (E-value = 2.4 e-2), which suggested a specific association between HEC1 and the promoter of auxin responsive genes (*Hosoda et al., 2002*) (*Figure 7B*). To investigate the relevance of these interactions for meristem regulation, we analysed the promoters of all ARF and Aux-IAA factors expressed in the SAM (*Figure 7C*; *Figure 7—figure supplement 2*; *Supplementary file 1*) (*Vernoux et al., 2011*). Strikingly, we found that HEC1 bound to 23 out of 25 promoters of auxin

signalling components known to be active in the SAM, a rate significantly higher than expected by chance (*Figure 7C*; *Figure 7—figure supplement 2*). Consistently, we observed that G-boxes and AREs found in HEC1 binding regions were significantly more closely spaced than expected from their relative positions across the whole genome (*Figure 7D*). Along these lines, we found that the most frequently occurring distance between G-Boxes and AREs in HEC1 binding regions was less than 50 bp and in immediate proximity to the peak summit (*Figure 7D*; *Figure 7—figure supplement 1E*). Examples for a promoter that exhibited such a close distance of HEC and ARF binding sites included the regulatory region of the auxin receptor *TRANSPORT INHIBITOR RESPONSE 1* (*TIR1*) (*Gray et al., 2001*), (*Figure 7—figure supplement 1F*). Furthermore, we found a significant enrichment of HEC1-regulated genes carrying an ARE and a G-box in their promoter (*Figure 7—figure supplement 1G–I*). Taken together, these results indicated that HEC1 and ARFs bind to the same genomic regions, either in competition, independently, or as a complex.

Given the strong but slow negative effect of HEC1 on auxin transcriptional output, we hypothesised that HEC1 could interfere with the positive feedback in auxin signalling (*Bhatia et al., 2016*) either by binding site competition or direct physical interaction with ARFs. To test this hypothesis, we analysed the potential for interaction between HEC transcription factors and MP, the key ARF orchestrating primordia initiation at the SAM (*Yamaguchi et al., 2013*). Both Yeast-2-Hybrid and Bimolecular-Fluorescence Complementation assays robustly demonstrated a physical interaction between MP and HEC1, HEC2, or HEC3, respectively and thus suggested that HEC factors could act as transcriptional modifiers for ARF activity (*Figure 8A–B*) (*Simonini et al., 2016*). In line with these results, HEC1 and HEC2, as well as their key cofactor SPATULA (SPT) were able to physically interact with BRAHMA in Yeast-two-Hybrid assays (*Figure 8C*) (*Efroni et al., 2013*). Taken together, these results showed that HEC transcription factors genomically associate with ARF targets and suggested that in the context of the SAM, HECs might be part of a higher order protein complex that modulates MP activity.

Given the slow repressive activity of HEC1 on auxin signalling and its association with MP, we next investigated what regulatory changes could mediate its impact on the auxin feedback system. To test this, we performed RNA-seq on micro-dissected shoot apices 14 hr after dex induction of *p16:HEC1-linker-GR* plants. In contrast to the set of early targets, we found that transcript levels of essential components for primordia initiation, such as MP itself, but also PID, a kinase required for proper polar localisation of PIN auxin transporters, were substantially reduced in this dataset (*Figure 8D*). In line with this finding, we observed a dramatic decrease in MP-GFP protein accumulation at the boundary zone after HEC1 induction in this domain, demonstrating the relevance of this interaction in the SAM (*Figure 8E–F*; *Figure 8—figure supplement 1A*). Importantly, the expression of the YUCCA auxin biosynthetic genes required for flower primordia formation was not affected by HEC1, suggesting that HEC factors likely do not work via the modulation of auxin production (*Supplementary file 1*; *Cheng et al., 2006*). Consistent with the decrease of MP-GFP accumulation and the global collapse of auxin output observed by DR5v2, we found that PIN1-GFP polarity was severely disturbed after *HEC1* induction (*Figure 8—figure supplement 1C–E*). In contrast, PIN1-GFP expression levels remained stable or even increased (*Figure 8—figure supplement 1C–E*; *Bhatia et al., 2016*). Consistently, the activity of the R2D2 auxin input sensor was also changed and domains of low auxin perception, usually restricted to the boundaries of the SAM, expanded substantially towards the centre of the SAM over time (*Figure 8G–H*; *Figure 8—figure supplement 1B*). To test whether these dramatic alterations in auxin signalling caused by *HEC1* also translated into stable modifications of cell fate, we analysed the expression of the boundary zone marker *CUC2* after *HEC1* stimulation, since the boundary is marked by a small, but stable local auxin minimum (*Bhatia et al., 2016*; *Heisler et al., 2005*). In line with the idea that *HEC1* potently interfered with cell fate decisions at the periphery by disruption of localized auxin signalling, we found a massive expansion of *pCUC2:3xGFP-NLS* expression (*Figure 8I–J*).

The slow effect of HEC1 on the auxin input and output patterns in the SAM and the absence of auxin biosynthetic and signalling components among the direct targets were consistent with the hypothesis that HEC1 may interfere with auxin signalling via its positive feedback loop (*Bhatia et al., 2016*). To test this more directly, we aimed at establishing epistasis between the auxin signalling system and HEC activity by using recent evidence that auxin can promote the expression of MP and PIN1, which in turn leads to a stabilization of the signalling system (*Bhatia et al., 2016*; *Heisler et al., 2005*). Therefore, we stimulated auxin signalling by chemical treatment while at the

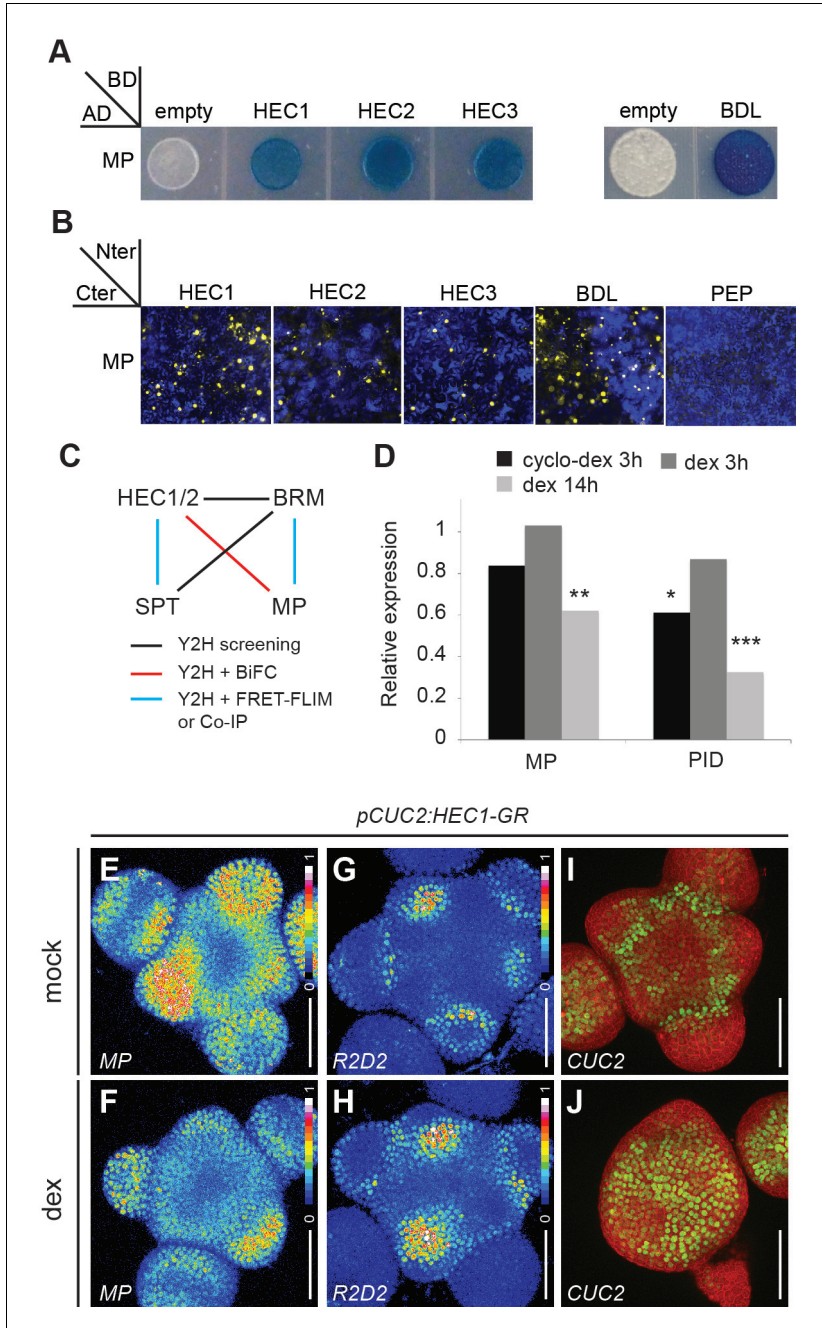

**Figure 8.** HEC factors interact with MP and unbalance the auxin feedback system. (**A**) Yeast-two-Hybrid assay between MONOPTEROS (MP) and HEC1, HEC2 or HEC3. Empty vector: negative control; BODENLOS (BDL): positive control. Blue staining denotes physical interaction. (**B**) Bimolecular-Fluorescence Complementation assay between MP and HEC1, HEC2 or HEC3. Nuclear YFP fluorescence was reconstituted in all combinations. PEP: negative control; BDL: positive control. (**C**) Protein-protein interaction network between HEC, SPT, MP and BRM. (**D**) Relative expression of *MP* and *PID* after *p16:HEC1-linker-GR* induction as measured by RNA-seq experiment. Biological triplicates were analysed. (**E,F**) Representative expression of *pMP:MP-GFP* one day after mock (**E**) or dex (**F**) treatment of *pCUC2:HEC1-linker-GR* plants (n > 4 per condition, intensity based colour coding). (**G, H**) Representative expression of *R2D2* one day after mock (**G**) or dex (**H**) treatment of *pCUC2:HEC1-linker-GR* plants (n > 4 per condition, intensity based colour coding). (**I–J**) Representative expression of *pCUC2:3xGFP-NLS* three days after mock (**I**) or dex (**J**) treatment of *pCUC2:HEC1-linker-GR* plants (n > 3 per conditions). Statistical test: Fischer´s exact test (EdgeR), *p<0.05, **p<0.01, ***p<0.001. Scale bar: 50 µm.

DOI: https://doi.org/10.7554/eLife.30135.033

*Figure 8 continued on next page*

*Figure 8 continued*

The following figure supplement is available for figure 8:

**Figure supplement 1.** Dynamics of MP-GFP, PIN1-GFP and R2D2 after HEC1-GR induction at the PZ/BZ.

DOI: https://doi.org/10.7554/eLife.30135.034

same time boosting *HEC1* expression and scored for expression of a key auxin signalling component, as well as for SAM phenotypes over time. In line with our RNA-seq data, we observed a significant reduction of *MP* expression 24 hr after HEC1 induction (*Figure 9A*). However, co-treatment with auxin rescued MP mRNA expression levels, indicating that our approach to stabilize the auxin feedback system was successful (*Figure 9A*). We next analysed the phenotypic outcome of this double perturbation. Strikingly, although *pCUC2:HEC1-linker-GR* and *p16:HEC1-linker-GR* inductions alone inhibited primordia initiation, auxin co-treatment suppressed this phenotype (*Figure 9C–J*, *Figure 9—figure supplement 1E*). Importantly, this suppression was neither the result of reduction of the key cofactor SPT, nor from the inhibition of *CUC2* promoter activity driving HEC1-linker-GR, further supporting that auxin acts downstream of HEC function during primordia initiation (*Figure 9B*; *Figure 9—figure supplement 1A–D*).

Together, these results demonstrated that HEC function is able to control the cell fate switch from peripheral meristematic cell to organ cell identity by locally interfering with auxin signalling, likely via its feedback system.

## *HECATE* genes integrate environmental signals to adjust SAM homeostasis

Having delineated a core regulatory system for controlling the timing of cell fate transitions in the shoot meristem, we wondered about the developmental relevance for this layer of control. Since plant cell fate is strictly determined by position, the timing of cellular transitions is intrinsic to the system under stable and optimal growth conditions. However, under changing environments, the regulatory system needs to adapt the morphogenetic output to the available resources, while at the same time conserving the functional pattern of the SAM (reviewed in [*Pfeiffer et al., 2017*]). Thus, we hypothesized that *HEC* genes could contribute to the modulation of SAM activity and growth in response to the environment. To test this hypothesis, we challenged the nutritional status of wild type and *hec1,2,3* mutants by shifting plants for 14 days to low light conditions (15 µmol m$^{-2}$ s$^{-1}$) just after bolting and assessed developmental responses at the SAM (*Jones et al., 2017*).

In line with previous studies, we observed that wild type plants displayed a substantially smaller SAM under low light conditions (*Jones et al., 2017*), and additionally observed a three-fold reduction in cytokinin responses compared to plants grown under normal light intensity (*Figure 10A,C,E, F,H,J*). Importantly, this reduction in the SAM size did not result from changes in cell size, suggesting that meristematic activity was reduced in these plants (*Figure 10—figure supplement 1A–F*). In contrast to wild type, we did not observe significant changes in the size of SAMs in *hec1,2,3* plants under low light, demonstrating that *HEC* genes are required for SAM adaptation to environmental changes (*Figure 10B,D,E*). Interestingly, even in *hec1,2,3*, cytokinin signalling was responsive to nutritional status and we observed a decreased TCSn reporter activity similar to wild type plants (*Figure 10F–J*). This suggested that cytokinin signalling and SAM size might be uncoupled in the *hec1,2,3* mutants. Taken together, this experiment revealed that *HEC* genes may have a key role during SAM adaptation to environmental challenges and suggested that regulating the timing of cell fate transitions might be important for developmental plasticity.

## Discussion

Cellular fate decisions occurring at the shoot apical meristem have important implications for the establishment and maintenance of plant architecture. Using precise spatio-temporal perturbations of gene expression, quantitative live-cell imaging and computational modelling, we revealed that HECATE bHLH transcription factors modulate cell fate transitions and coordinate the dynamics of cell fate decisions across key developmental domains of the SAM by balancing cytokinin and auxin phytohormonal signals (*Figure 11*).

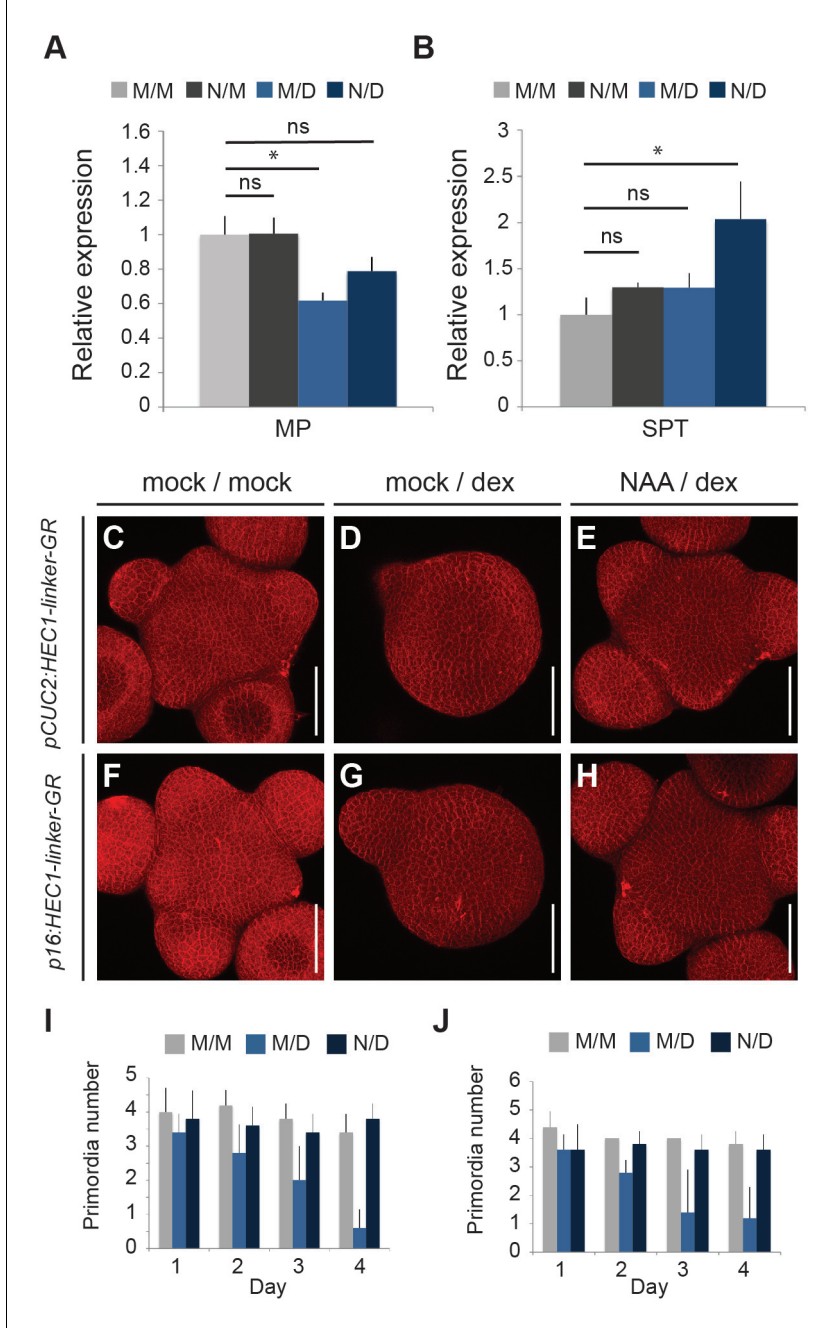

**Figure 9.** Auxin feedback stabilization suppresses HEC function at the SAM periphery. (**A–B**) Relative expression of *MP* (**A**) and *SPT* (**B**) 24 hr after *p16:HEC1-linker-GR* induction (M/M: mock/mock, N/M: NAA/mock, M/D: mock/dex, N/D: NAA/dex) (**C–E**) Representative views of *pCUC2:HEC-linker-GR* SAMs four days after mock/mock (**C**) mock/dex (**D**) or NAA/dex (**E**) treatment. (**F–H**) Representative view of *p16:HEC-linker-GR* SAMs four days after mock/mock (**F**) mock/dex (**G**) or NAA/dex (**H**) treatment. (**I**) Time series quantification of primordia number in *pCUC2:HEC1-linker-GR* after mock/mock (M/M), mock/dex (M/D) or NAA/dex (N/D) treatment (n = 5 per condition and time point). (**J**) Time series quantification of primordia number in *p16:HEC1-linker-GR* after mock/mock (M/M), mock/dex (M/D) or NAA/dex (N/D) treatment over time (n = 5 per condition and time point). Statistical test: Welch t-test *p<0.05, **p<0.01, ***p<0.001. Scale bar: 50 μm.

DOI: https://doi.org/10.7554/eLife.30135.035

The following source data and figure supplement are available for figure 9:

**Source data 1.** qRT-PCR (panel 9A and B);
DOI: https://doi.org/10.7554/eLife.30135.037

*Figure 9 continued on next page*

*Figure 9 continued*

**Figure supplement 1.** Dynamics of primordia initiation after HEC1-GR induction at the PZ/BZ and auxin co-treatment.
DOI: https://doi.org/10.7554/eLife.30135.036

In contrast to our earlier findings where using steady state end-point phenotypes we concluded that HEC function act partially independently of the WUS/CLV3 system and repress cytokinin

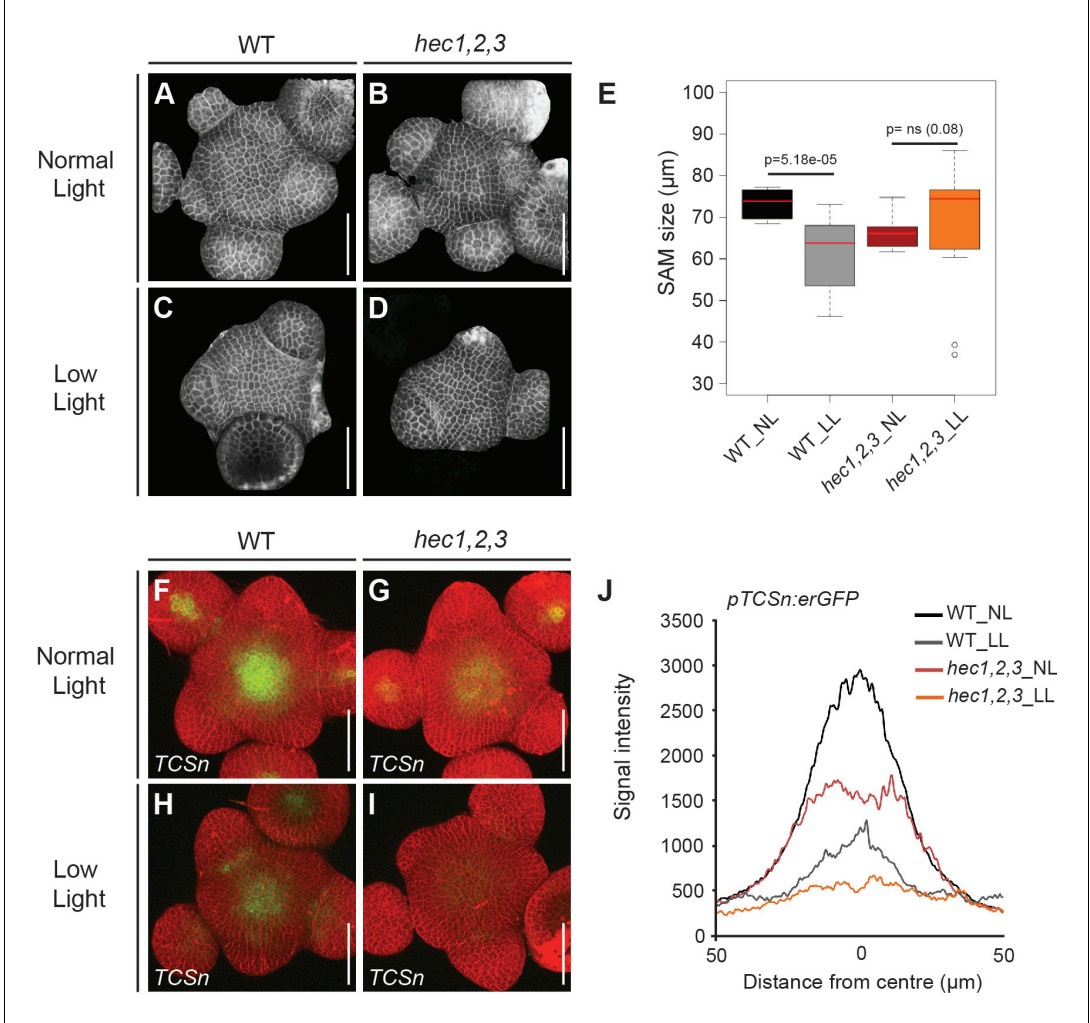

**Figure 10.** HEC function integrates environmental signals at the SAM. (**A–D**) Reconstructed views of shoot meristems in wild type (**A, C**) and *hec1,2,3* (**B, D**) plants grown under normal (**A, B**) and low light intensity conditions (**C, D**) (n > 13). (**E**) Shoot meristem size in wild type and *hec1,2,3* plants grown under normal and low light intensity regimes. (**F–I**) Cytokinin response (*pTCSn:erGFP*) in shoot meristems of wild type (n > 6) (**F, H**) and *hec1,2,3* (n > 4) (**G, I**) plants under normal (**F, G**) and low light intensity conditions (**H, I**). (**J**) Quantification of cytokinin response in wild type and *hec1,2,3* shoot meristem under normal and low light regimes. Statistical test: Wilcoxon signed-ranked test (**E**). Scale bar: 50 μm.
DOI: https://doi.org/10.7554/eLife.30135.038

The following source data and figure supplement are available for figure 10:

**Source data 1.** Source data provided for panel E and intensity plot profiles *pTCSn:erGFP* under different light regimes (panel J).
DOI: https://doi.org/10.7554/eLife.30135.040

**Source data 2.** Cell surface area MorphographX (*Figure 10—figure supplement 1E*);
DOI: https://doi.org/10.7554/eLife.30135.041

**Figure supplement 1.** Cell size at the SAM after transfer to low light conditions.
DOI: https://doi.org/10.7554/eLife.30135.039

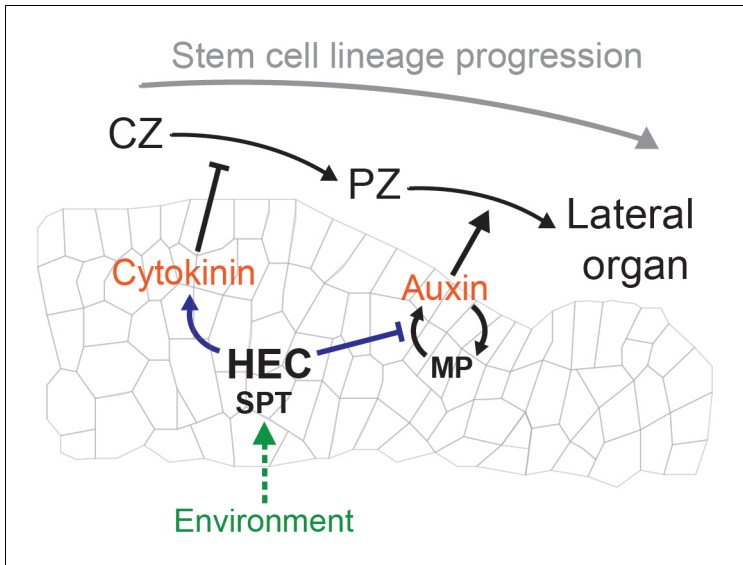

**Figure 11.** Theoretical model depicting HEC activity controlling cellular fate transition in the SAM by modulating the auxin-cytokinin phytohormonal balance and integrating environmental signals.
DOI: https://doi.org/10.7554/eLife.30135.042

signalling by promoting type-A ARRs expression (*Schuster et al., 2014*), we now show that *HEC* function in the centre of the SAM not only interferes with the core *WUS/CLV3* regulatory system, but also promotes the enlargement of the OC, CZ and the cytokinin signalling domain. These contradicting results can be reconciled by considering the dynamics of HEC function. Early HEC activity promotes CLV3, WUS and cytokinin domain expansion and consequently elevates type-A ARR expression, which are primary targets of cytokinin signalling (*Bhargava et al., 2013*). In turn, A-type ARRs act as negative regulators of cytokinin signalling and constitute a negative feedback dampening *WUS* expression (*Schuster et al., 2014*). Furthermore, the additive regulation of SAM size by *HEC* and *WUS* function (*Schuster et al., 2014*), together with the ability of HEC1 to ectopically promote cytokinin signalling in the root meristem, where *WUS* is not expressed, suggests that *HEC* function primarily acts on cytokinin signalling at the SAM and in turn promotes *WUS* expression. However, given the indirect regulation of both *WUS* and cytokinin, it will be important in future studies to further clarify the network topology and to identify the intermediate regulatory components mediating *HEC* regulatory function.

These results highlighted the power of time resolved analyses coupled to transient perturbations in studying the SAM, which now allowed us to discriminate direct from indirect effects arising as a consequence of feedback mechanisms. Since traditional loss-of-function mutations are inherently stable and thus do not lend themselves to this type of approach, we employed computational modelling to test divergent regulatory scenarios, which could not be analysed experimentally. The model not only allowed us to identify processes that were sufficient to explain the experimental observations, but also helped to rule out alternative hypotheses, such as stem cell re-specification, for which simulations could not reproduce in vivo data. Modelling and experimentation suggested that the combination of a reduced stem cell system with increased organ initiation rate observed in *hec1,2,3* triple mutants likely was caused by a faster differentiation of stem cells. This occurred at least two levels, namely an increased rate of primordia initiation, as well as a reduced time for the primordium to grow and separate from the SAM.

While the regulation of cell fate transition dynamics has not received much attention in the plant field with the exception of the stomatal lineage (*Simmons and Bergmann, 2016*), a large body of work from animal model systems has addressed this issue (*Busch et al., 2015*; *Maduro, 2010*; *Marciniak-Czochra et al., 2009*). It has emerged that cells within a developmental lineage undergo specific phenotypic steps on their trajectory towards terminal differentiation, however whether fate decisions occur deterministically or rather stochastically is still unresolved and might strongly depend

on the cell type and the developmental context (*Moris et al., 2016*). Given the purely position-dependent fate regulation observed in plant shoots, cells do not progress along a deterministic cell fate trajectory, but rather acquire alternative identities until they reach their final position in an organ and fully differentiate accordingly. Consequently, the transition from stem cell to transit-amplifying cell and further on to primordium founder cell mainly pertains to a timing of differentiation rather than providing intrinsic information on final cell fate. We have found that HEC transcription factors act in accordance with this idea and modulate the relative speed of the successive cell fate transitions at the SAM rather than specifying a specific developmental outcome.

Previous theoretical studies on plant stem cell systems have focused on pattern formation (*Chickarmane et al., 2012*; *Espinosa-Soto et al., 2004*; *Robinson et al., 2011b*; *Yadav et al., 2013*), morphogenesis (*Kierzkowski et al., 2012*; *Kuchen et al., 2012*) or cell division behaviour (*Louveaux et al., 2016*), and only few studies investigated how local signals can coordinate the growth of different functional domains (*Grieneisen et al., 2007*; *Mähönen et al., 2014*). While lacking 3-dimensional resolution for the sake of simplicity, our 2D cell population model has allowed us to provide a theoretical framework on how fate decision events are coordinated along stem cell differentiation trajectories and how affecting key transition checkpoints during this process quantitatively modulates dynamics of the stem cell system.

In line with a multi-step model, we previously showed that *HEC1* is expressed in all relevant domains of the SAM and that its expression is under direct control of WUS (*Figure 2—figure supplement 1A*) (*Schuster et al., 2014*). Thus, in addition to specifying stem cell fate, WUS may play an additional role in facilitating the transition from CZ to PZ fate via transcriptional repression of *HEC1* (*Schuster et al., 2014*).

Using local modulation of HEC activity, and given the low HEC1 protein mobility in the SAM, we also revealed that HEC factors do not promote cell proliferation locally but rather non-cell autonomously (*Daum et al., 2014*; *Schuster et al., 2014*). Interestingly, these changes in cell behaviour were reminiscent of the increased PZ cell number observed after local perturbation of the *WUS/CLV3* feedback system in the CZ (*Reddy and Meyerowitz, 2005*; *Yadav et al., 2010*). Although the mechanisms responsible for the communication between CZ and PZ are still unresolved, the convergence of *HEC* and *WUS* function in controlling cytokinin signalling, and its role in promoting cell cycle progression (*Riou-Khamlichi et al., 1999*), points towards a potential function of cytokinin signalling in mediating this inter-domain communication.

In addition to their role in regulating cytokinin, we found that HEC proteins modulate the auxin regulatory loop. One mechanism, which could be responsible for this effect, could be the physical interaction with the auxin response factor MP. MP plays a central role in stabilizing the auxin feedback system via non-cell autonomous control of PIN1 polarity towards the site of MP accumulation (*Bhatia et al., 2016*). This self-reinforcing regulatory system dynamically builds auxin maxima and generates sites of high MP accumulation which subsequently trigger a switch to primordia fate (*Bhatia et al., 2016*). Importantly, protein-protein interaction data suggest that HEC1, HEC2 and SPT physically interact with the SWI/SNF chromatin remodelling ATPase BRAHMA (BRM), which also operates in a protein complex with MP during primordia initiation (*Figure 8C*; *Wu et al., 2015*). Although the mechanistic details of their interaction still remain elusive, we propose that the HEC-SPT complex could modify MP-BRM function by direct physical interaction and thereby could modulate the dynamics of the entire auxin feedback system. Consistently, a reduction of *HEC* function would result in increased MP-BRM activity, which in turn enhances the auxin feedback system to instruct the initiation of flower primordia at a higher rate. Alternatively, HEC-SPT complex could indirectly regulate the expression of key components of the auxin feedback system, independently of the physical interaction with MP or locally reduce the levels of available auxin. Both models could explain the fairly slow, progressive changes observed in the dynamics of auxin perception, transport and response after promoting *HEC* activity at the periphery of the SAM. It will therefore be important in the future to mechanistically dissect the function of the physical interaction between HEC factors and MP-BRM complex to further reveal how HEC function impacts on the auxin feedback dynamics.

Similarly to the shoot stem cell system, the balance between auxin and cytokinin is essential to control the dynamics of stem cell differentiation at the root apical meristem (*Di Mambro et al., 2017*; *Dello Ioio et al., 2008*). Our finding that HEC function can ectopically shift this hormonal balance and can impact on the dynamics of RAM differentiation suggests that plant cells can read out

hormonal inputs and integrate this information to specify their identity along their differentiation trajectory. Along the same lines, auxin and cytokinin are essential for several aspects of cambial activity, including restriction of stem cell fate, cambial cell proliferation, and xylem differentiation (*Bhalerao and Fischer, 2014*; *Brackmann et al., 2017*; *Hejatko et al., 2009*). However, in this context, the exact function of individual hormones on the progression of cell fate acquisition and the nature of their interaction still remains elusive.

In addition to the integration of hormonal and transcriptional signals to control cell fate decisions, the SAM adjusts its activity in response to environmental signals including light or nutritional status (*Pfeiffer et al., 2016*; *Jones et al., 2017*). Although this dynamic process is crucial to understand the molecular basis for plant developmental plasticity, the regulatory network mediating SAM homeostasis remains poorly characterized. Our findings that HEC function is required to adjust SAM size in response to low light suggests that it defines a regulatory hub in integrating environmental cues at the SAM. It will be important to further characterize the molecular mechanism underpinning this response and to unravel how HEC function interact with light signalling components during SAM activity (*Zhu et al., 2016*). Furthermore, it will be important in the future to systematically test the role of known stem cell regulators and assess their regulatory function in the SAM upon various environmental challenges including temperature, nutrients, light intensity or biotic interactions. These experiments could reveal the mechanisms of developmental plasticity and how the regulatory landscape of the shoot stem cell system adjusts and rewires in response to the environment.

# Materials and methods

## Cloning

*pCLV3:HEC1-linker-GR*, *pCUC2:HEC1-linker-GR* and *p16:HEC1-linker-GR* constructs were generated by ligation of HEC1 coding sequence (CDS) with a 33 aa Serine-Glycin linker and GR tag into pDONOR221 vector and recombined in pGreenIIs constructs (*Schuster et al., 2014*). *pKNOLLE:fast-mFluorescentTimer-NLS* was generated from fast m-FluorescentTimer CDS fused to N7 NLS and introduced by subsequent BP and LR reactions (Thermo Fischer Scientist, Waltham, Massachusetts, USA) in a pGreenIIs vector containing 2.1 kb of genomic sequence upstream of the *KNOLLE* start codon. The CUC2 promoter used correspond to the 3.2 kb genomic sequence upstream the ATG. *pCLV3:HEC1-linker-GFP*, *pCUC2:HEC1-linker-GFP*, *pCUC2:3xGFP-NLS* and *p35S:HEC1-linker-GFP* were cloned using the Green Gate system (*Lampropoulos et al., 2013*). N7-NLS was used as NLS tag (*Daum et al., 2014*).

For Yeast-two-Hybrid and Bi-Fluorescence complementation (BiFC) assay, HEC1, HEC2, HEC3, MP, BDL and PEP CDS were PCR amplified using Phusion Taq-polymerase (New England Biolabs, Inc., Massachusetts, USA) and subsequently cloned by Gibson assembly in pGILDA/pB42AD (Yeast-two-Hybrid) (*Gibson, 2011*) or ligated in pGreenII0179 (SPYCE constructs) or pGreenII0229 (SPYNE cassettes) via NotI (BiFC complementation) (*Rodríguez-Cazorla et al., 2015*; *Waadt et al., 2008*)

## Primers

A detailed list of primers used in this study can be found in *Supplementary file 2*.

## Creation transgenic lines

Plants were transformed with *Agrobacterium tumefaciens* ASE strain by floral dipping according to standard protocols. All HEC1-GR inducible lines are homozygous for a single T-DNA insertion and were subsequently crossed to the corresponding reporter lines.

## Plant material

*pWUS:3xYFP-NLS_pCLV3:mCherry-NLS* (*Pfeiffer et al., 2016*), *pTCSn:erGFP* (*Zürcher et al., 2013*), *pPIN1:PIN1-GFP* (*Heisler et al., 2005*), *hec1,2,3* (*Schuster et al., 2014*), *pDR5v2:3xYFP-NLS*, R2D2 (*Liao et al., 2015*), *wus-1_pWUS:WUS-GFP* (*Daum et al., 2014*) were previously described. *hec1,2,3* were PCR genotyped as described in (*Schuster et al., 2014*).

## Plant growth and treatments

All plant lines generated in this study are in the Col-0 background. Plants were grown at 23°C, 65% humidity under long day conditions (16 hr light/8 hr dark) with LED lights or white lights (Philips, Amsterdam, Netherlands) at approximately 200 µmol m$^{-2}$ s$^{-1}$.

For the light shift experiments, plants were grown under white light at 200 µmol m$^{-2}$ s$^{-1}$ until flowering transition. As soon as the first flower primordia were observed, plants were transferred to low light intensity regimes (approximately 15 µmol m$^{-2}$ s$^{-1}$) and kept for 14 days before imaging.

For dexamethasone treatment (Sigma_D4903, St. Louis, Missouri, United States) of the shoot apices, a solution of 10 µM dex, 0.01% ethanol and 0.015% Silwet were manually sprayed and applied on top of the inflorescence meristem of 25–30 DAG plants. Mock treatment (0.01% ethanol and 0.015% Silwet) was conducted similarly. For 1-Naphtalenacetic acid (NAA, Sigma, St. Louis, Missouri, United States) treatment, 1 mM was applied according to previous studies (*Heisler et al., 2005*). Shoot apices were treated once at the first day of the experiment.

Cytokinin treatment was performed by treating inflorescences with a solution of 50 µM 6-Benzylaminopurine (BA, Sigma, St. Louis, Missouri, United States) supplemented with 0.015% Sylwet once every 5 days. Inflorescence meristems were analysed after 8 days.

The cumulated silique number was measured by counting over time the total number of flower above stage 15 emerging after plant bolting. The inflorescence plastochron was then obtained by calculating the average time separating the emergence of 2 successive siliques. The number of flower primordia at a given time point were counted up to flower stage 2.

For root meristem analysis, plants were grown vertically on 0.5x MS (Duchefa, Haarlem, The Netherlands), 0.8% Phytoagar plates under long day conditions (16 hr light/8 hr dark) with white lights (Philips, Amsterdam, Netherlands). 3DAG seedlings were subsequently transferred and grown vertically on media supplemented with ethanol (0.01%) and with or without dex (10 µM).

## Yeast-two-Hybrid assay

The EGY48 yeast strain (-Ura) was cotransformed with the combination of pGilda and pB42AD constructs under study or with the corresponding empty vector or negative control. Positive colonies were selected on solid media (–Ura, -His, -Trp + glucose) and verified by PCR. Induction for testing protein-protein association and colorimetric assays on plates were assayed as decribed in (*Ripoll et al., 2015*)

## Bi-Fluorescence complementation assay

Empty versions of pGreen0179 + SPYCE or pGreenII0229 + SPYNE were used as controls and T-DNA binary vectors were transformed into the *Agrobacterium* strain AGL-0. *Nicotiana benthamiana* leaves were infected and YFP fluorescence assayed 72 hr after inoculation under a Nikon Eclipse TE2000-U epifluorescence microscope. The reciprocal assays for all the BiFC interactions shown in this study were performed obtaining the same results as presented in *Figure 7A–B* (data not shown). Tobacco leaves used for these assays were also co-injected with the *Agrobacterium* strain expressing the viral suppressor p19 (*Voinnet et al., 2003*).

## Chromatin immuno-precipitation (ChIP-seq)

For Chromatin immuno-precipitation followed by sequencing, we used functional homozygous *p35S: HEC1-linker-GFP* in the *hec1* background and performed individual ChIP as previously described (*Schuster et al., 2015*). Individual samples were then pooled (10 to 12 individual ChIP/replicate) by precipitation with 50 µl NaAc, 10 µl Acrylamid and 1 ml ethanol and incubated at −80°C for overnight. Samples were then centrifuged for 1 hr at 4°C and air-dried in a sterile bench before being resuspended in sterile water. Biological duplicates were analysed. Raw data has been deposited at NCBI GEO under the series number GSE94311.

## Expression analysis: RNA-seq and qRT-PCR

RNA extraction was performed using the RNAeasy plant mini kit according to manufacturer instructions (Qiagen, Hilden, Germany), on individually micro-dissected and pooled inflorescence meristems (till flower stage 3–4; 15–20/replicate).

For qRT-PCR, cDNA was prepared using cDNA synthesis kit after DNase treatment (Thermo Fischer Scientist, Waltham, Massachusetts, USA), qPCR was performed using SYBR Green kit (EurX, Gdańsk, Poland).

For RNA-seq and Chip-seq, libraries and next-generation sequencing were performed according to standard protocols (core sequencing facility, Bioquant, Heidelberg University). For RNA-seq, biological triplicates were analysed. Raw data has been deposited at NCBI GEO under the series number GSE94311.

## Image acquisition and analysis

All confocal pictures were acquired using Nikon (Minato, Tokyo, Japan) A1 Confocal with a CFI Apo LWD 25x water immersion objective. For time series analysis, settings were established at the first day on mock samples and were kept during the course of the experiment. Shoot meristems were manually dissected by cutting of the stem, removing the flowers and were counterstained with 1 mg/ml DAPI. Root meristems were counterstained with propidium iodide (Sigma, St. Louis, Missouri, United States) at 0,1 mg/ml. Individual populations of 5 to 15 plants were analyzed daily.

To quantify *pWUS:3xYFP-NLS*, *pCLV3:mCherry-NLS*, *pTCS:erGFP* intensity plot profiles, we used Fiji software (*Schindelin et al., 2012*). Z-stacks were first averaged using gaussian blur, and used to create a maximum projection picture. Intensity plot profiles were then generated using a ROI (line) crossing the SAM on its centre (*Figure 2—figure supplement 3*). To quantify WUS, CLV3 and TCS domain size, intensity plot profiles from maximum projection pictures previously generated were analysed. The size of the domain was obtained by measuring the distance including all points displaying an intensity value higher than one quarter of the maximum intensity value.

SAM size was measured using the Nikon A1 software, by averaging three diameter segments starting from primordia 1 (P1), P2 and P3 and crossing the meristem at its centre.

For quantifying the number of cells in the SAM, pre-processing, segmentation and data analysis was done using a customized workflow for the KNIME Image Processing platform (KNIP) (*Berthold et al., 2008*). 3D visualization of analysed image stacks was done using the Fiji 3D viewer (*Schindelin et al., 2012*). To obtain overall cell numbers a ubiquitously expressed reporter (*pUBQ10:3xGFP-NLS*) was used for imaging. To count numbers of CLV3-expressing cells, this reporter construct was combined with a respective stem cell-specific transcriptional reporter *pCLV3:mCherry-NLS*.

For image processing, meristem image volumes were background subtracted and segmented by a 3D seeded watershed algorithm provided by the KNIP package (*Berthold et al., 2008*). Using different pre-processing the same image stack was used to create a 3D mask for the whole meristem using again 3D seeded watershed. Borders between meristem and emerging flower primordia were marked manually to obtain a 3D mask that was used to filter out nuclei residing outside of the inflorescence meristem.

To quantify numbers of CLV3-expressing stem cells nuclear segments from the *pUBQ10:3xmCherry-NLS* channel were used to obtain mean intensities in the CLV3 channels. Cells were considered to be positive for CLV3 if their respective mean nuclear intensity was higher than 35% of the maximum mean value.

To analyse cell proliferation, a cell cycle-regulated transcriptional reporter was constructed (*pKNOLLE:fast-mFluorescentTimer-NLS*) and combined with a ubiquitously expressed reporter (*pUBQ10:3xGFP-NLS*). Nuclear segments were obtained as described before and mean intensities for the timer (blue channel) and the GFP channel were measured for each segment. The ratio between the blue and the green mean intensities were normalized and assigned to four different classes (Class1: 0–0.3, Class2: 0.3–0.53, Class3: 0.53–0.76, Class4: 0.76–1). Cells with the highest ratio (Class4) represent young cells with a very recent cell division (<4 hr, data not shown).

To compare proliferation rates in the inner (central) and outer (peripheral) domain of the meristem a sphere of radius r was fitted through the centroids of the L1 cells of the meristem summit (all L1 cells with a distance of <= 35 µm from a manually selected center point P of the meristem) using a Matlab function. To adjust the size of the inner (central) domain to overall meristem size all cells with a distance to P smaller then 0.33 * r were considered to belong to this central domain (defined from the size of CLV3 domain in other plant lines), whereas cells with a distance larger then 0.33 * r were classified as peripheral cells.

Root cortex cell number was quantified as described in (*Dello Ioio et al., 2007*). Cumulative cortex cell length was quantified as described in (*Kang et al., 2017*).

## Bioinformatic analysis

The quality of the sequence files quality was first confirmed using FastQC (http://www.bioinformatics.babraham.ac.uk/projects/fastqc).

For RNA-seq analysis, read alignment and peak calling was then performed using TOPHAT2 algorithms using default settings (*Kim et al., 2013*). BAM files were converted to SAM files using samtools and read tables were constructed using HT-seq (*Anders et al., 2015*; *Li et al., 2009*). Next, individual HT-seq tables were combined in a common table and differential gene expression was calculated using EdgeR with p<0,05 as a cut-off for differentially expressed genes (*Robinson et al., 2010*) (*Supplementary file 1*).

For ChIP-seq analysis, reads were aligned using BOWTIE2 algorithms using default settings and peak calling was performed with macs2 (*Langmead and Salzberg, 2012*; *Zhang et al., 2008*). During peak calling, we limited the number of duplicated reads to 2 ('keepdup 2'). Next, 200 bp regions were defined around the peak summits and the overlapping intervals of the two biological replicates were intersected using bedtools "multiinter" function (*Quinlan and Hall, 2010*). The overlapping peaks were then annotated using Homer and used to locate the peaks in relation to gene model (*Heinz et al., 2010*) (*Supplementary file 1*). ChIP peaks were visualised using the Integrative Genomic Viewer (IGV) (*Robinson et al., 2011a*).

For de novo motif identification, a 500 bp region around the overlapping peak summits was defined and used for de novo motif discovery using MEME-ChIP with JASPAR core 2016 as motif input (*Bailey et al., 2009*).

For comparing DNA-binding regions of HEC1, SPCH (*Lau et al., 2014*), PIF3, PIF5 (*Pfeiffer et al., 2014*), KAN1 (*Merelo et al., 2013*) and LFY (*Moyroud et al., 2011*), 50 bp region were centered on peak summits. Regions with 80% or more overlap were defined as shared binding regions.

To obtain the position of G-boxes and ARE across the genome, bed files were generated using IGV (*Robinson et al., 2011a*). To measure the relative distribution between G-box and ARE on the entire genome, the closest ARE for each G-box were detected using only open chromatin regions (*Zhang et al., 2012*) and distances were reported using the bedtools 'closest' function with –d option. To measure their distribution under HEC1 binding regions, G-boxes under HEC1 peaks were identified using bedtools 'intersect' function. For each G-box, the closest ARE in open chromatin was next detected (*Zhang et al., 2012*). Distances were reported using bedtools 'closest' function and histograms were constructed using R (https://www.r-project.org/).

To assess the percentage of HEC1 target genes carrying G-box, ARE or both and the distribution of these motifs in open chromatin regions, respective bed files were first used to generate lists of annotated genes using the homer function 'annotate.peaks.pl'. List of genes regulated by HEC1 (p<0.05) were next used and intersected with the list of genes carrying G-box, ARE or both in open chromatin regions using Microsoft Excel (*Heinz et al., 2010*; *Zhang et al., 2012*). Significance tests were performed using two-sided Fisher's exact test.

## Computational modelling

We developed a cell population model that takes into account the simplified geometry of the SAM. The model describes the evolution in time of different SAM structures, i.e., CZ, incipient primordia and unspecified PZ cells. Cells are continuously displaced towards the periphery by the divisions of cells more centrally located, with cells located at the outer boundary of the CZ transiting to PZ, depending on local WUS concentrations. Incipient primordia are initiated near the central boundary of the PZ and primordia separate from the meristem at fixed time after their initiation. PZ cells that do not contribute to primordia contribute to longitudinal growth of the plant. The following key processes were considered: (i) proliferation of cells in CZ and PZ, (ii) fate transition from the CZ to the PZ and from the PZ to the organs, (iii) initiation of incipient primordia, (iv) separation of primordia from the meristem and (v) contribution of meristem cells to longitudinal growth.

Different feedbacks were included in the model. As suggested by our in vivo data, the initiation frequency of incipient primordia and the transition from the CZ to the PZ, both depend on the CZ

cell number. The model is formulated using a system of ordinary differential equations, which number may vary in time. A detailed description of the model is found in *Supplementary file 3*.

## Acknowledgements

We thank Che-Yang Liao and Dolf Weijers for sharing R2D2 and DR5v2 reporters before publication, David Ibbersson and the CellNetworks sequencing core facility for conducting next-generation sequencing and Christoph Schuster for helpful discussions.

## Additional information

### Funding

| Funder | Grant reference number | Author |
| --- | --- | --- |
| Deutsche Forschungsge-meinschaft | SFB873 | Anna Marciniak-Czochra Jan U Lohmann |
| European Social Fund | Elite program for postdocs | Anne Pfeiffer |
| Baden-Württemberg Stiftung | Elite program for postdocs | Anne Pfeiffer |
| National Institutes of Health | 1R01GM112976-01A1 | Martin F Yanofsky |

The funders had no role in study design, data collection and interpretation, or the decision to submit the work for publication.

### Author contributions

Christophe Gaillochet, Conceptualization, Data curation, Formal analysis, Investigation, Visualization, Writing—original draft, Writing—review and editing; Thomas Stiehl, Data curation, Software, Investigation, Writing—original draft, Writing—review and editing; Christian Wenzl, Lanxin Li, Anne Pfeiffer, Jana P Hakenjos, Joachim Forner, Resources, Investigation; Juan-José Ripoll, Formal analysis, Investigation, Visualization; Lindsay J Bailey-Steinitz, Investigation, Visualization; Andrej Miotk, Formal analysis, Investigation; Martin F Yanofsky, Resources, Supervision, Funding acquisition; Anna Marciniak-Czochra, Data curation, Supervision, Funding acquisition; Jan U Lohmann, Conceptualization, Supervision, Funding acquisition, Methodology, Writing—original draft, Writing—review and editing

### Author ORCIDs

Christophe Gaillochet http://orcid.org/0000-0003-0537-1356
Thomas Stiehl http://orcid.org/0000-0001-9686-9197
Juan-José Ripoll http://orcid.org/0000-0002-8229-1555
Anne Pfeiffer http://orcid.org/0000-0001-6825-6297
Andrej Miotk https://orcid.org/0000-0003-2581-672X
Joachim Forner https://orcid.org/0000-0002-6406-7066
Anna Marciniak-Czochra http://orcid.org/0000-0002-5831-6505
Jan U Lohmann http://orcid.org/0000-0003-3667-187X

### Decision letter and Author response

Decision letter https://doi.org/10.7554/eLife.30135.048
Author response https://doi.org/10.7554/eLife.30135.049

## Additional files

### Supplementary files

• Supplementary file 1: ChIPseq, RNAseq and QRT-PCR analyses
DOI: https://doi.org/10.7554/eLife.30135.043
• Supplementary file 2: List of oligonucleotides

DOI: https://doi.org/10.7554/eLife.30135.044
• Supplementary file 3: Description of computational model
DOI: https://doi.org/10.7554/eLife.30135.045

### Major datasets

The following dataset was generated:

| Author(s) | Year | Dataset title | Dataset URL | Database, license, and accessibility information |
|-----------|------|---------------|-------------|--------------------------------------------------|
| Gaillochet C, Lohmann JU | 2017 | Control of plant cell fate transitions by transcriptional and hormonal signals | https://www.ncbi.nlm.nih.gov/geo/query/acc.cgi?acc=GSE94311 | Publicly available at the NCBI Gene Expression Omnibus (accession no. GSE94311) |

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
