## [Decision Letter]

[Editors’ note: a previous version of this study was rejected after peer review, but the authors submitted for reconsideration. The first decision letter after peer review is shown below.]

Thank you for submitting your work entitled "Control of plant cell fate transitions by transcriptional and hormonal signals" for consideration by *eLife*. Your article has been reviewed by three peer reviewers, one of whom is a member of our Board of Reviewing Editors, and the evaluation has been overseen by a Senior Editor.

Our decision has been reached after consultation between the reviewers. Based on these discussions and the individual reviews below, we regret to inform you that this version of your work will not be considered further for publication in *eLife*.

The reviewers thought the work was well written and interesting, and the experiments performed were of high technical quality. However, all three felt that current data are not sufficient to draw the conclusions made and that alternative interpretations could lead to other explanatory models. The reviewers have each made a number of suggestions for improvement and their detailed reviews are appended below.

In summary, there are several concerns that would need to be addressed to make this suitable for publication in *eLife*, but if the authors can provide experimental data to address these issues, the editor and reviewers strongly encourage submission of an updated manuscript. As this is likely to take more than the 2 months typical for "revision", a new submission would be required, but the editors will be made aware that the submission was encouraged.

Priority concerns

1) The "cell-non autonomous" claim for HEC1 needs to be backed up with experimental evidence appropriate for the tissues being monitored in this study (see reviewer 2 comment 1).

2) Many claims are made using only gain of function/ectopic expression experiments that are sufficient to show that HEC family members can interfere with a normal process, but not that they normally control it. Experiments done in mutant backgrounds or by reducing HEC expression in tissue specific ways were suggested by all reviewers.

3) Work presented here contradicts previous work (Schuster et al., 2014) that concluded that HEC1 inhibits CK signaling during control of the SAM. Although the data concerning CK signaling in this new manuscript are convincing, the authors should discuss this (and other) discrepancies with previous work.

4) The mechanism of "HEC factors directly modulate auxin signal transduction by physical interaction with MONOPTEROUS" is overstated and only one such possible way that HEC1 could influence auxin. Additional data supporting this mode and discussing alternatives are needed.

5) More functional evidence for the connection between HEC, auxin and cytokinin is needed, as well as better modeling.

Reviewer #1:

In this well written and interesting article, the authors go beyond current models of stem-cell maintenance in the *Arabidopsis* shoot meristem and investigate the transitions cells make from slow dividing behaviors in the CZ of meristem out through the peripheral zone (PZ) which is more of a transit amplifying cell population, and into developing organ primordia. Starting with the counterintuitive phenotype of smaller CZ but faster organ initiation in *hec1/2/3* mutants, they use live imaging and modeling to describe the paths cells take along this transition, and manipulate the system to investigate the roles of well-known hormones (Auxin and cytokinin) and regulatory factors (WUS/CLV3) in the process. They also include a clever "timer" protein to see how quickly cells divide during manipulations. To my knowledge this hasn't been used in plants before, but I think it would be a valuable tool for the community.

Priority concerns

A) "HEC function appeared not only required and sufficient to control stem cell to PZ, but also mediated PZ to organ fate transitions" This line of reasoning was based on ectopic expression of HEC in the PZ, which says that it can interfere with a normal process, but not that it controls it. I would like to see an amiRNA (or other LOF) experiment to substantiate this conclusion. Also, the expression patterns of HEC1/2/3 are never described in this manuscript. I imagine since they cite 4 prior publications, the patterns are known. This information would be useful included in this manuscript (at least in text). The same issue hold true for the root experiments. Induction of HEC1 can perturb the system, but does it actually work here?

B) Reporters for CK and auxin are considered to be evidence for direct regulation of these pathways. However, I wonder whether these reporters might be very downstream. For example, could pTCS-GFP signal report that cells are in a proliferative state and have high hormone response because of that state, rather than the other way around? To really nail the idea that HEC1/2/3 work through auxin or cytokinin, wouldn't doing these overexpression experiment in a background mutant for ARRs or ARFs be more direct?

C) The ChIP-seq is done on overexpressed HEC1 and the targets include typical bHLH sites. Could this just be generic binding? It would be nice to take the published bHLH ChIP-seq datasets (PIFs, POPEYE, SPEECHLESS) and see that the HEC1 sites are not always overlapping with any bHLH.

D) I think modeling can be a useful addition to developmental studies and was used well in Figure 4 to describe how *hec1/2/3* might affect a cellular behavior to result in the observed SAM phenotypes. Such models could be very helpful in testing the proposed auxin signaling feedback loops, as I felt this part of the manuscript was veering into "just so stories" about how things could work, but that alternative models could also explain the data.

Reviewer #2:

This manuscript by Gaillochet et al. describes the function of the HECATE1 (HEC1) gene in regulating cell fate in the *Arabidopsis* shoot meristem. The authors perform experiments to demonstrate that HEC1 both inhibit the progression from Central Zone (CZ) identity to Peripheral Zone (PZ) identity and subsequently inhibits the transition from PZ to primordium. The experiments have been carried out using cell-type specific inducible expression of HEC1 in both the CZ domain (under CLV3 promoter) and in the Boundary Zone (BZ) (under CUC2 promoter). The authors next move to investigate the interaction with auxin and cytokinin (CK) hormone dynamics and propose a model for this. In parallel they analyse HEC1 ChIP-seq and RNA-Seq data to reveal potential target genes and identify the Auxin Response Factor MONOPTEROUS (M) as a potential HEC1 interacting factor.

There are several interesting aspects of the work, which is also technically well-performed. However, it seems to me that the current data are not sufficient to draw the conclusions that the authors do and that alternative interpretations could lead to other models to explain them.

Specific issues:

1) It is throughout the manuscript assumed that HEC1 can mediate its effect non-cell autonomously. This is based on the observation in Schuster et al., 2014, where the authors show that HEC1 expressed in the CZ inhibits WUS expression in the OZ via repression of CK signaling. The situation in this manuscript is different: firstly, the cells where HEC1 is supposed to function non-cell autonomously are not the OZ, but the PZ and secondly in Schuster et al. concludes that HEC1 repress CK signaling whereas here they demonstrate that HEC1 stimulate CK signaling (see further comment on this below). In order for the authors to make the non-cell autonomous claim, they will need to create an immobile version (e.g. HEC1-3xGFP) and show the effects are still there.

2) The authors list three scenarios for how HEC1 can increase the cell number of the meristem and use elegant inducible systems in specific domains of the SAM. This part is very complicated and I am not convinced about the authors' interpretation. For example, they conclude that HEC1 does not increase stem cell division locally (scenario 1). In order to draw such a conclusion, they should carry out the analysis in a zone-specific manner (as opposed to counting the entire SAM, which will mask local differences) and within 2 days, since CLV3 is downregulated at 4 days. Moreover, to eliminate scenario 2 they argue that the constant ratio of CLV3 positive cells over total SAM cells means that PZ cells are not re-specified to stem cells. In order to make this conclusion, they need a marker for PZ and show that this would not 'turn off' upon induction of HEC1 activity. Finally, based on the elimination of scenarios 1 and 2 (which I am not convinced about), they conclude that scenario 3 must be correct without providing experimental evidence for this, but suggest it is supported by showing that the subsequent transition from PZ to primordia is regulated similarly.

3) Generally, the experiments using ectopic inducible expression of HEC1 under cell-type specific promoters would be much stronger if they were done in the *hec1/2/3* background to test what aspects of this triple mutant is rescued (or not) by the transgene. Without this, it is hard to conclude what the effect of HEC1 would be on these processes in a WT situation.

4) In the Abstract and in the text, it is stated that data in this manuscript "show that HEC factors directly modulate auxin signal transduction by physical interaction with MONOPTEROUS […]" This overstates what the experiments actually show and more work would be required to validate this. The authors have shown that 1) MP is one among 12 ARF genes that are expressed in the SAM and which are direct targets of HEC1, 2) that many HEC1 targets contain both G-boxes (predicted HEC1 recognition sites) and Auxin Response Elements (AREs) and 3) that HEC1 can interact with MP in yeast and in BiFC. However, in order to make a convincing case, they need to provide genetic evidence for this interaction and establish if HEC1 actually does modify MP-mediated auxin signaling. Moreover, there are 12 other relevant ARFs according to the authors (i.e. expressed in the SAM). Does HEC1 also interact with them? Modify their activity?

5) The authors aim to address whether HEC genes can control cell fate transition in other developmental contexts, such as the root apical meristem (RAM) and leaves, through the regulation of auxin/cytokinin crosstalk. The conclusions that the authors draw from their experiments are again overstated, since they only analysed response to CK signaling in *p16:HEC1-linker-GR* roots with no attempt to show how auxin signaling response is affected. At the same time, the authors linked the leaf phenotype of the ectopic HEC1 expression to changes in auxin signaling without any prove nor attempting to understand possible crosstalk with cytokinin. The following lists issues particularly regarding this part:

– It's not clear whether HEC1 governs meristem size by controlling CK signalling or whether it is the result of a feedback mechanism operated by auxin. How (when and where) is the auxin response affected after HEC1 induction?

– It would be helpful if the author could better describe in which zones/tissues p16 is expressed in the RAM? From Figure 6—figure supplement 3, it seems *p16:mCherry-linker-GR-NLS* is expressed in the division zone of the root meristem. Does *p16:mCherry-linker-GR-NLS* mark the stem cell niche and/or the transition zone?

– The control of CK on root meristem size occurs only from the vascular tissue at the transition zone. Is the activity of HEC1 sufficient to regulate root meristem size operating from the transition zone (could use the RCH2 promoter to address this)? Does the TCS expression increase before the meristem size starts diminishing?

– Generally, 16hrs of CK treatments are sufficient to shrink the meristem. To address whether HEC1 controls root meristem size by recruiting cytokinin signalling, two days of DEX treatment is too long. Moreover, the description of how the authors treated their roots in these experiments is unclear; in the text it is stated "one day after induction of *p16:HEC1-linker-GR* we observed a reduction of RAM size, which correlated with increased cytokinin signaling at the transition zone" referring to Figure 6—figure supplement 3. However, the legend to this figure says DEX treatment was for two days.

– From Figure 6—figure supplement 3 the ectopic periclinal divisions within the cortical layer are not very clear. Please provide a better magnification of it.

– In the text "HEC1 induction in seedlings (Figure 6—figure supplement 3)" should refer to 3I-L. In addition, panels K and L are out of focus and it is not possible to appreciate the phenotypes that are described.

6) The conclusion from a previous paper from this group (Schuster et al., 2014) concluded that HEC1 inhibits CK signaling during control of the SAM by inducing ARR7/15. Here, they describe the opposite. Although the data concerning CK signaling in this new manuscript are convincing, it would be desirable if the authors could discuss this discrepancy.

7) It is concluded that induction of HEC1 leads to PIN1 polarization defects referring to Figure 7 and Figure 8—figure supplement 1. I would agree that the signal of PIN1-GFP is stronger, but it is not possible from these images to make conclusions on localization. Moreover, even if polarization was compromised, it is possible that this could be due to over-accumulation of PIN1 in these cells and not an effect on HEC1 on PIN polarity as such.

Reviewer #3:

In this manuscript Gaillochet and colleagues address the interesting question of how cell fate transitions are coordinated in meristems. Utilizing a combination of computational/genetic/molecular approaches they show that a family of BHLH transcription factors called HECATE (HEC) is central in this coordination. Indeed, they demonstrate that HEC transcription factors regulate cell non-autonomously cell fate transition from stem cells to transit amplifying cells and from transit amplifying to organ primordia. Subsequently, they investigate whether HEC might regulate cell transition via phyto-hormones and they focus on cytokinin and auxin. Via analysis of *TCSn::erGFP* and *DR5v2:3xYFP* in loss and gain of functions of HEC they show that HEC induces cytokinin activity and represses the auxin one. They also demonstrate that HECs can control cell fate transition via auxin and cytokinin in the root as expression of HEC1 in p16 domain decreases the root meristem length and increases cytokinin activity. To conclude, they show that HEC1 binds auxin responsive elements and physically interacts with ARF5 and BRAHMA to modulate auxin signalling.

I really enjoyed reading Gaillochet et al. manuscript, nevertheless the connection of HEC with hormones looks to me not extremely robust, as functional evidences were not reported. I could not understand whether the modulation of auxin and cytokinin is necessary and sufficient to mediate HEC function and this point should be address to validate their model.

Also, in previous report from the same group (Schuster et al., 2014) it was shown that HEC1 promotes type A-ARR expression, negative regulators of cytokinin activity. The authors should discuss with care this point in the discussion.

[Editors’ note: what now follows is the decision letter after the authors submitted for further consideration.]

Thank you for resubmitting your work entitled "Control of plant cell fate transitions by transcriptional and hormonal signals" for further consideration at *eLife*. Your revised article has been favorably evaluated by Christian Hardtke (Senior editor), three original reviewers (one of whom is a member of our Board of Reviewing Editors), and one new reviewer.

The manuscript has been improved but there are some remaining issues that need to be addressed before acceptance, as outlined below:

Overall the reviewers found this to be an engaging read and an interesting conceptual advance. Some softening of bold statements (e.g. hormone dependency claims somewhat oversold based on the actual evidence) and a bit more testing of how HEC could modulate transitions in a non-mutant condition, would make this a valuable addition to the plant development literature. Below are 4 essential issues to address in a revision, and in a response to reviewers.

1) A critical issue is whether HEC part of the normal regulatory circuit controlling meristem size, and is hard to really be sure of this when using mutants and mis-expression. Recently, however, there were some interesting studies (Murray lab, Nature Communications) about cell cycle length and cell size in the meristem where changing nutritional status or light allowed them to make meristems of different sizes. If HEC is part of a homeostatic mechanism, then there should be some clear predictions of reporter/mutant behavior under these different conditions. These experiments would not require any new materials, but could be very helpful to define HECs normal function.

2) We urge caution in interpreting the effects of HEC expression in the RAM by looking just at root length, as length can be changed by altering cell behaviors in different ways.

Several more precise methods are commonly used to capture the kinetics of transition from proliferation to expansion, and one or more of these should be employed. For example, (1) evaluation of cell division frequency at the root meristems of *P16::HEC-GR* seedlings when compared to wt, or (2) to score the number of cells from QC to first differentiated cell as described (e.g. Perilli and Sabatini Plant Developmental Biology 2010 pp 177-187), and/or progression of cell size (e.g. Kang, Breda and Hardtke, 2017).

Once this finer dissection of the phenotype is performed, you should reconsider whether HEC has different effects in shoot and root or not.

3) The distinction between HEC-direct and WUS-mediated effects on CK is not clear. Reviewers were not sure that the pathway as presented is sufficiently supported by the current evidence, In particular, how can HEC direct versus WUS mediated effect on the CK signaling be distinguished? In the *hec* mutant, loss of HEC activity might result in higher CLV3 expression, which suppresses WUS. Attenuation of WUS would lead to high ARR7, ARR15 expression and suppression of CK signalling. In the *CLV3::HEC -GR* enhanced HEC activity correlates with increase of WUS expression which when followed by attenuation of ARR7, ARR15 expression ultimately enhances CK signaling. Hence I am not sure one can exclude that CK signaling (monitored by TCS) is under WUS control.

To address this, in the PZ zone, the effect of HEC on CK could be measured without the complication of WUS by visualizing the TCS reporter in this background.

4) The finding of G-boxes and AREs in common regions is not terribly surprising given the high frequency of these motifs in the genome. It would add substance to this section if (1) the global analysis [1]was filtered for euchromatic regions only. Since HEC1 peaks are 75% within 3 KB of genes, they are probably more in euchromatin than heterochromatin. An entire genome comparison therefore would be skewed-best to mask the heterochromatic regions when doing this analysis. Detailed promoter analysis might be considered out of the scope of this article the behavior of dual motif-containing genes should be monitored in the HEC-GR induction data (reviewer 3) to see if the correlation holds. This would provide additional support for relevance of the model.

---

## [Author Response]

[Editors’ note: the author responses to the first round of peer review follow.]

Reviewer #1:[…] Priority concernsA) "HEC function appeared not only required and sufficient to control stem cell to PZ, but also.mediated PZ to organ fate transitions" This line of reasoning was based on ectopic expression of HEC in the PZ, which says that it can interfere with a normal process, but not that it controls it. I would like to see an amiRNA (or other LOF) experiment to substantiate this conclusion.

We fully agree that domain-specific knock-down experiments would be the optimal strategy to investigate the role of HEC factors in controlling cellular fate transitions. Therefore, we previously worked hard on trying to use domain-specific MIGS or amiRNA, but given the difficulty to knock down simultaneously *HEC1, HEC2* and *HEC3*, these approaches were unfortunately not successful. Furthermore, we found that stem cell specific expression of MIGS against mCherry caused silencing in the entire shoot meristem, precluding the use of this system to create true domain-specific loss-of-function. To better fit the presented data, we modified the text accordingly.

Also, the expression patterns of HEC1/2/3 are never described in this manuscript. I imagine since they cite 4 prior publications, the patterns are known. This information would be useful included in this manuscript (at least in text). The same issue hold true for the root experiments. Induction of HEC1 can perturb the system, but does it actually work here?

We have clarified HEC expression patterns by including a schematic and citing relevant previous publications. We have also included gene expression data from the root meristem, showing that HEC genes are not expressed in this tissue.

B) Reporters for CK and auxin are considered to be evidence for direct regulation of these pathways. However, I wonder whether these reporters might be very downstream. For example, could pTCS-GFP signal report that cells are in a proliferative state and have high hormone response because of that state, rather than the other way around? To really nail the idea that HEC1/2/3 work through auxin or cytokinin, wouldn't doing these overexpression experiment in a background mutant for ARRs or ARFs be more direct?

We mostly used reporters as readout for hormonal signalling and do not propose a detailed mechanism from their analysis, we further clarified this in the text. Regarding the TCS reporter, we showed that stem cell specific induction of HEC1 reduces the mitotic index at the centre of the SAM while promoting cytokinin signalling in this domain, excluding that this reporter would merely report the proliferative state of those cells.

We also agree that it would be important to further uncover the molecular players acting in these hormonal pathways downstream of HEC1. To bridge this gap, we mostly focused our analysis of the interaction between HEC and MP by showing that HEC1 indirectly represses MP expression, leading to reduced MP-GFP accumulation and pin-like inflorescences. Unfortunately, the experiments overexpressing HEC in the mutants interfering with cytokinin signalling, such as *arr1,10,12*, or *35S:CKX, log septuple,* or *ahk2,3,4* are not possible, since all promoters for SAM specific expression strongly respond to cytokinin signalling status (we have tested p16 and pCLV3).

C) The ChIP-seq is done on overexpressed HEC1 and the targets include typical bHLH sites. Could this just be generic binding? It would be nice to take the published bHLH ChIP-seq datasets (PIFs, POPEYE, SPEECHLESS) and see that the HEC1 sites are not always overlapping with any bHLH.

We analysed the overlap between HEC1 binding peaks and those obtained from three bHLH and two structurally unrelated transcription factors. We observe that a large proportion of peaks are not shared between these bHLH factors, demonstrating the specificity of HEC1 DNA-binding pattern.

D) I think modeling can be a useful addition to developmental studies and was used well in Figure 4 to describe how hec1/2/3 might affect a cellular behavior to result in the observed SAM phenotypes. Such models could be very helpful in testing the proposed auxin signaling feedback loops, as I felt this part of the manuscript was veering into "just so stories" about how things could work, but that alternative models could also explain the data.

We are happy that the reviewer found the combination of experimental and modelling approaches useful to understand HEC loss-of-function phenotypes. We also agree that using a model that would integrate gene networks could greatly help in testing multiple regulatory scenarios. However, although our current model can be used to phenomenologically describe cell population dynamics, its structure does not allow us to implement a complex gene-regulatory network. We feel that building a new model including gene regulatory networks would go beyond the scope of this manuscript.

Reviewer #2:[…] There are several interesting aspects of the work, which is also technically well-performed. However, it seems to me that the current data are not sufficient to draw the conclusions that the authors do and that alternative interpretations could lead to other models to explain them.

We are happy that the reviewer found our work interesting and technically well executed. We have carried out more experiments to support our current model and further discussed alternative hypothesis.

Specific issues:1) It is throughout the manuscript assumed that HEC1 can mediate its effect non-cell autonomously. This is based on the observation in Schuster et al., 2014, where the authors show that HEC1 expressed in the CZ inhibits WUS expression in the OZ via repression of CK signaling. The situation in this manuscript is different: firstly, the cells where HEC1 is supposed to function non-cell autonomously are not the OZ, but the PZ and secondly in Schuster et al. concludes that HEC1 repress CK signaling whereas here they demonstrate that HEC1 stimulate CK signaling (see further comment on this below). In order for the authors to make the non-cell autonomous claim, they will need to create an immobile version (e.g. HEC1-3xGFP) and show the effects are still there.

We fully agree that using HEC1-3xGFP fusions would allow us to prove the non cellautonomous activity of HEC1 and we thank the reviewer for this suggestion. However, we previously prepared HEC1-GFP-GR and GR-GFP-HEC1 proteins that proved to be nonfunctional, as increasing protein size interfered with HEC1 functionality. Unfortunately, this technical limitation would make the result of the suggested experiment impossible to interpret. To this address this question, we chose an alternative approach and analysed HEC1-GFP protein mobility by driving its expression under various domains of the SAM. We previously published that HEC1-GFP was restricted to the WUS domain when driven under its promoter (Daum et al., 2014) and fully cell-autonomous when driven from the pML1 promoter (Schuster et al., 2014). Consistently, we now find that *pCUC2:HEC1-linker-GFP* and *pCLV3:HEC1-linker-GFP* in *pCLV3:3xmCherry-NLS*, we observe that HEC1-GFP protein does not move long distance laterally, supporting our statement that HEC function in the CZ mostly acts on the PZ cell non-autonomously.

2) The authors list three scenarios for how HEC1 can increase the cell number of the meristem and use elegant inducible systems in specific domains of the SAM. This part is very complicated and I am not convinced about the authors' interpretation. For example, they conclude that HEC1 does not increase stem cell division locally (scenario 1). In order to draw such a conclusion, they should carry out the analysis in a zone-specific manner (as opposed to counting the entire SAM, which will mask local differences) and within 2 days, since CLV3 is downregulated at 4 days. Moreover, to eliminate scenario 2 they argue that the constant ratio of CLV3 positive cells over total SAM cells means that PZ cells are not re-specified to stem cells. In order to make this conclusion, they need a marker for PZ and show that this would not 'turn off' upon induction of HEC1 activity. Finally, based on the elimination of scenarios 1 and 2 (which I am not convinced about), they conclude that scenario 3 must be correct without providing experimental evidence for this, but suggest it is supported by showing that the subsequent transition from PZ to primordia is regulated similarly.

We added experimental data and changed the text accordingly to support and clarify our imaging analysis. First, we quantified the mitotic index in the centre and at the periphery of the SAM independently, as suggested by the reviewer. In line with our previous observations, we found that HEC function does not locally promote cell division in the CZ. To analyse early PZ cell re-specification, the reviewer suggested analysing the expression of a PZ marker after HEC induction. From our experience, no such reporter clearly delimits CZ from PZ domains, making this suggested experiment impossible to perform (we previously tried the UFO promoter which displayed a patchier expression pattern than expected). Therefore, we analysed the ratio between the number of CLV3 cells and the total number of cells, which should increase (especially in the 2 first days of induction) if PZ de-differentiation would take place. By showing no dramatic changes in this value, we excluded this scenario. Importantly, computational modelling simulations were used to test all the results from those experiments.

3) Generally, the experiments using ectopic inducible expression of HEC1 under cell-type specific promoters would be much stronger if they were done in the hec1/2/3 background to test what aspects of this triple mutant is rescued (or not) by the transgene. Without this, it is hard to conclude what the effect of HEC1 would be on these processes in a WT situation.

We have now applied domain specific rescue by *pCUC2:HEC1-linker-GFP* in *hec1,2,3* mutant background and observed similar phenotypes than in the WT background, supporting our described function of HEC factors within distinct domains of the SAM. This result was in line with our previous experiments using *pCLV3:HEC1* in hec1,2,3 mutants, which showed SAM expansion comparable to WT background (Schuster et al., 2014).

4) In the Abstract and in the text, it is stated that data in this manuscript "show that HEC factors directly modulate auxin signal transduction by physical interaction with MONOPTEROUS […]" This overstates what the experiments actually show and more work would be required to validate this. The authors have shown that 1) MP is one among 12 ARF genes that are expressed in the SAM and which are direct targets of HEC1, 2) that many HEC1 targets contain both G-boxes (predicted HEC1 recognition sites) and Auxin Response Elements (AREs) and 3) that HEC1 can interact with MP in yeast and in BiFC. However, in order to make a convincing case, they need to provide genetic evidence for this interaction and establish if HEC1 actually does modify MP-mediated auxin signaling. Moreover, there are 12 other relevant ARFs according to the authors (i.e. expressed in the SAM). Does HEC1 also interact with them? Modify their activity?

We thank the reviewer for this comment and now better discuss alternative hypothesis regarding HEC mode of action in regulating auxin responses in the Discussion section. Given the similar pin-like inflorescence phenotypes of *mp* mutants and *pCUC2:HEC1* mutants and that MP is the only ARFs that we found significantly regulated after 14 hours of HEC1 induction, we focused on characterizing the interaction between HEC and MP during the initiation of primordia. However, we agree that additional ARFs could play a role. We now show that HEC1 can reduce MP-GFP levels at the PZ and that HEC activity represses *MP* expression at the RNA level. We have also included new data that show that this repression can be rescued by auxin application, demonstrating that HEC function acts by modulating *MP* expression and auxin signalling during primordia initiation.

5) The authors aim to address whether HEC genes can control cell fate transition in other developmental contexts, such as the root apical meristem (RAM) and leaves, through the regulation of auxin/cytokinin crosstalk. The conclusions that the authors draw from their experiments are again overstated, since they only analysed response to CK signaling in p16:HEC1-linker-GR roots with no attempt to show how auxin signaling response is affected. At the same time, the authors linked the leaf phenotype of the ectopic HEC1 expression to changes in auxin signaling without any prove nor attempting to understand possible crosstalk with cytokinin. The following lists issues particularly regarding this part:

We included information on HEC activity in the root meristem and now show that these factors are not expressed in this region. Therefore, our analysis of *p16:HEC1-linker-GR* roots exclusively aims at testing whether HEC function is sufficient to regulate cytokinin/auxin responses and in turn impact cell fate acquisition in another developmental context, rather than characterizing in depth HEC function at the root meristem. We further clarified this in the text and removed data presented at the seedling stage to avoid driving the reader away from the main focus of our manuscript.

– It's not clear whether HEC1 governs meristem size by controlling CK signalling or whether it is the result of a feedback mechanism operated by auxin. How (when and where) is the auxin response affected after HEC1 induction?

We now analysed DR5v2 24 hours after induction and found a reduced intensity at the transition zone, demonstrating that HEC1 can act on auxin and cytokinin signalling in the root. Given the absence of endogenous HEC activity in the root, we feel that determining the chain of events would contribute little to the main story of our manuscript.

– It would be helpful if the author could better describe in which zones/tissues p16 is expressed in the RAM? From Figure 6—figure supplement 3, it seems p16:mCherry-linker-GR-NLS is expressed in the division zone of the root meristem. Does p16:mCherry-linker-GR-NLS mark the stem cell niche and/or the transition zone?

We modified the figure accordingly and have marked the transition zone.

– The control of CK on root meristem size occurs only from the vascular tissue at the transition zone. Is the activity of HEC1 sufficient to regulate root meristem size operating from the transition zone (could use the RCH2 promoter to address this)? Does the TCS expression increase before the meristem size starts diminishing?

Although it would be interesting to understand the fine spatio-temporal dynamics of HEC1 activity in the root meristem, we feel that this experiment would contribute little to understanding how HEC1 acts in the SAM, where it is expressed. We have used the root only to test whether the effects on auxin and cytokinin could be transferred to a tissue devoid of endogenous HEC activity.

– Generally, 16hrs of CK treatments are sufficient to shrink the meristem. To address whether HEC1 controls root meristem size by recruiting cytokinin signalling, two days of DEX treatment is too long. Moreover, the description of how the authors treated their roots in these experiments is unclear; in the text it is stated "one day after induction of p16:HEC1-linker-GR we observed a reduction of RAM size, which correlated with increased cytokinin signaling at the transition zone" referring to Figure 6—figure supplement 3. However, the legend to this figure says DEX treatment was for two days.

We included pictures of meristems 24 hours after HEC induction.

– From Figure 6—figure supplement 3 the ectopic periclinal divisions within the cortical layer are not very clear. Please provide a better magnification of it.

Good point, we included a magnified picture of the ectopic periclinal division.

– In the text "HEC1 induction in seedlings (Figure 6—figure supplement 3)" should refer to 11I-L. In addition, panels K and L are out of focus and it is not possible to appreciate the phenotypes that are described.

For reasons outlined above, we felt that this experiment was driving the reader away from the core message of our manuscript and for clarity we decided to remove it.

6) The conclusion from a previous paper from this group (Schuster et al., 2014) concluded that HEC1 inhibits CK signaling during control of the SAM by inducing ARR7/15. Here, they describe the opposite. Although the data concerning CK signaling in this new manuscript are convincing, it would be desirable if the authors could discuss this discrepancy.

We thank the reviewer for this comment and further explain the causes of this discrepancy in the Discussion section (short term vs. long term effects).

7) It is concluded that induction of HEC1 leads to PIN1 polarization defects referring to Figure 7 and Figure 8—figure supplement 1. I would agree that the signal of PIN1-GFP is stronger, but it is not possible from these images to make conclusions on localization. Moreover, even if polarization was compromised, it is possible that this could be due to over-accumulation of PIN1 in these cells and not an effect on HEC1 on PIN polarity as such.

We included magnified pictures to better observe the defects in PIN1 polar localization after HEC1 induction.

Reviewer #3:[…] I really enjoyed reading Gaillochet et al. manuscript, nevertheless the connection of HEC with hormones looks to me not extremely robust, as functional evidences were not reported. I could not understand whether the modulation of auxin and cytokinin is necessary and sufficient to mediate HEC function and this point should be address to validate their model.

We are delighted that the reviewer appreciated our work, and we have included new data to support the functional interaction between HEC factors and auxin and cytokinin signalling pathways.

Also, in previous report from the same group (Schuster et al., 2014) it was shown that HEC1 promotes type A-ARR expression, negative regulators of cytokinin activity. The authors should discuss with care this point in the discussion.

We agree that this is an important point and we have further clarified it in the Discussion section.

[Editors' note: the author responses to the re-review follow.]

[…] 1) A critical issue is whether HEC part of the normal regulatory circuit controlling meristem size, and is hard to really be sure of this when using mutants and mis-expression. Recently, however, there were some interesting studies (Murray lab, Nature Communications) about cell cycle length and cell size in the meristem where changing nutritional status or light allowed them to make meristems of different sizes. If HEC is part of a homeostatic mechanism, then there should be some clear predictions of reporter/mutant behavior under these different conditions. These experiments would not require any new materials, but could be very helpful to define HECs normal function.

We agree that studying how plants integrate environmental cues and relay these signals to stem cell systems are of high relevance to further define the mechanisms driving plant developmental plasticity. Surprisingly, the regulatory mechanisms driving SAM size adjustment and the regulatory function of key stem cell factors under various environmental cues remain poorly characterized, although some studies have paved the way by mostly describing the SAM developmental response at the cellular level (Landrein et al., 2015; Jones et al., 2017).

In this context, we tested whether HEC function could be part of the regulatory system controlling the adjustment of the SAM size to environmental perturbations by growing plants under low light conditions (15 µmol m^-2^ s^-1^) (Jones et al., 2017). By monitoring SAM size, we revealed that *hec1,2,3* mutants could not adjust their meristem in response to low light conditions, demonstrating the relevance of HEC function for SAM homeostasis. Interestingly, cytokinin signalling was still responsive to the low light treatment in the HEC loss-of-function mutants, suggesting a complex re-wiring of the regulatory system.

Although we feel that the fine dissection of the molecular mechanisms underpinning the SAM homeostatic response to low light inputs is certainly be an interesting topic for a follow-up study we feel that more details would go beyond the scope of our present manuscript. Thus, we further discussed the relevance of this approach in the Discussion section for future work in order to understand how the shoot stem cell system integrate various environmental signals–including temperature, light, nutrients or biotic interactions– and fine tune cellular fate transitions to adjust the SAM output.

2) We urge caution in interpreting the effects of HEC expression in the RAM by looking just at root length, as length can be changed by altering cell behaviors in different ways.Several more precise methods are commonly used to capture the kinetics of transition from proliferation to expansion, and one or more of these should be employed. For example, (1) evaluation of cell division frequency at the root meristems of P16::HEC-GR seedlings when compared to wt, or (2) to score the number of cells from QC to first differentiated cell as described (e.g. Perilli and Sabatini Plant Developmental Biology 2010 pp 177-187), and/or progression of cell size (e.g. Kang et al., 2017).Once this finer dissection of the phenotype is performed, you should reconsider whether HEC has different effects in shoot and roo, or not.

We thank the reviewers for their helpful comments. We have now further analyzed HEC activity at the cellular level in the RAM:

1) By measuring the number of cortex cells after HEC induction (Figure 6—figure supplement 3).

2) By calculating the cumulative cortex cell length 2 days after HEC induction

(Figure 6—figure supplement 3).

3) By qualitatively assessing cell division using the Fluorescent Timer (Figure 6—figure supplement 3).

Together with the shortening of the RAM length upon *p16:HEC1-linker-GR* induction, we observed that the number of cortex cells decreases, whereas neither cell length nor cell proliferation are altered in the relevant root regions. These data fully support the idea that HEC function would promote RAM cell differentiation by modulating the auxin/ cytokinin balance.

3) The distinction between HEC-direct and WUS-mediated effects on CK is not clear. Reviewers were not sure that the pathway as presented is sufficiently supported by the current evidence, In particular, how can HEC direct versus WUS mediated effect on the CK signaling be distinguished? In the hec mutant, loss of HEC activity might result in higher CLV3 expression, which suppresses WUS. Attenuation of WUS would lead to high ARR7, ARR15 expression and suppression of CK signalling. In the CLV3::HEC -GR enhanced HEC activity correlates with increase of WUS expression which when followed by attenuation of ARR7, ARR15 expression ultimately enhances CK signaling. Hence I am not sure one can exclude that CK signaling (monitored by TCS) is under WUS control.To address this, in the PZ zone, the effect of HEC on CK could be measured without the complication of WUS by visualizing the TCS reporter in this background.

We agree that given the lack of information on the directness of the interaction between HEC and WUS or cytokinin in our genomic data and given that WUS and cytokinin signaling cross-regulate in a positive feedback manner, both scenarios could explain our imaging data and thus required further clarification.

We further addressed this question by inducing *pCUC2:HEC1-linker-GR* and monitoring TCSn:erGFP (Figure 5—figure supplement 1). In contrast to cytokinin treatment, which promotes cytokinin signalling at the SAM periphery, inducing HEC function in this region did not. Although WUS could act downstream of HEC to mediate the domain-specific regulation of cytokinin signalling, other lines of evidence need to be considered.

First, HEC function represses MP expression at the SAM periphery, which in turn promotes cytokinin signalling by repressing type A ARRs (Zhao et al., 2010). Thus, HEC function indirectly represses CK at the periphery by down-regulating MP and by blocking the formation of lateral organs.

Second, we previously observed that *pCLV3:HEC1* and *clv3-7* genetically interact to produce larger shoot meristem than individual single mutants (Schuster et al., 2014). This argues against a full epistatic relation between HEC and WUS and thus indicates that HEC function acts partially independently of WUS to control SAM size. Along the same line, we did not observe PZ cell re-specification when inducing *pCLV3:HEC1-GR* as observed when inducing *WUS-GR* in the CLV3 domain(Yadav, Tavakkoli and Reddy, 2010), further dissociating HEC and WUS function in the SAM.

Finally, when ectopically expressed in the root meristem, HEC can promote cytokinin signalling in the absence of WUS (where it is not expressed), showing that in this context their function on cytokinin signalling can be uncoupled from WUS.

Together we think that these evidences support the idea that HEC function can promote cytokinin signalling partially independently of WUS. As our data cannot fully clarify the network topology between HEC, WUS and cytokinin, we toned down our conclusions and included this point in the Discussion section.

4) The finding of G-boxes and AREs in common regions is not terribly surprising given the high frequency of these motifs in the genome. It would add substance to this section if (1) the global analysis [1]was filtered for euchromatic regions only. Since HEC1 peaks are 75% within 3 KB of genes, they are probably more in euchromatin than heterochromatin. An entire genome comparison therefore would be skewed-best to mask the heterochromatic regions when doing this analysis. Detailed promoter analysis might be considered out of the scope of this article the behavior of dual motif-containing genes should be monitored in the HEC-GR induction data (reviewer 3) to see if the correlation holds. This would provide additional support for relevance of the model.

We fully agree with the reviewers, and apologize for the lack of clarity in the description of our bioinformatics approach. The presented data (Figure 7) was analyzed by filtering only for euchromatic regions (Zhang et al., 2012). We now further described our analysis in the Materials and methods section.

Additionally, we assessed the abundance of ARE and G-box or both motifs at the promoter of HEC1-response genes compared to the genome-wide scale (Figure 7—figure supplement 1). We found that the proportion of genes containing ARE and Gbox motifs on their promoter is significantly higher among HEC early target genes, (especially for CYC/DEX dataset) compared the genome-wide scale, supporting our model that HEC factors and MP regulatory functions converge at the promoter of target genes.

4) The finding of G-boxes and AREs in common regions is not terribly surprising given the high frequency of these motifs in the genome. It would add substance to this section if (1) the global analysis [1]was filtered for euchromatic regions only. Since HEC1 peaks are 75% within 3 KB of genes, they are probably more in euchromatin than heterochromatin. An entire genome comparison therefore would be skewed best to mask the heterochromatic regions when doing this analysis. Detailed promoter analysis might be considered out of the scope of this article the behavior of dual motif-containing genes should be monitored in the HEC-GR induction data (reviewer 3) to see if the correlation holds. This would provide additional support for relevance of the model.

We fully agree with the reviewers, and apologize for the lack of clarity in the description of our bioinformatics approach. The presented data (Figure 7) was

analyzed by filtering only for euchromatic regions (Zhang & al 2012, Plant Cell). We now further described our analysis in the method section.

Additionally, we assessed the abundance of ARE and G-box or both motifs at the

promoter of HEC1-response genes compared to the genome-wide scale (Figure 7—figure supplement 1). We found that the proportion of genes containing ARE and G-box motifs on their promoter is significantly higher among HEC early target genes, (especially for CYC/DEX dataset) compared the genome-wide scale, supporting our model that HEC factors and MP regulatory functions converge at the promoter of target genes.